# Galloylated liposomes enable targeted drug delivery by overcoming protein corona shielding

Jinbo Li[1,5], Jiang Yu[1,5], Jia Song[1], Yingxi Zhang[1], Ning Li[1], Zhaomeng Wang[2], Meng Qin[1], Mingming Zhao[1], Baoyue Zhang[1], Ruiping Huang[1], Shuang Zhou[1], Yubo Liu[1], Zhonggui He[1], Hongzhuo Liu [1,3] ✉, Dan Liu [4] ✉ & Yongjun Wang [1,3] ✉

Ligand-targeted nanomedicines provide precise delivery, enhance drug accumulation, and reduce side effects, but their clinical translation is hindered by challenges like protein corona formation, which can mask targeting ligands and impair functionality, and complex manufacturing processes. Here we develop galloylated liposomes (GA-lipo) by incorporating gallic acid-modified lipids into lipid bilayers, enabling the stable and controlled adsorption of targeting ligands through non-covalent physical interactions. This approach preserves ligand orientation and functionality, ensuring that binding sites remain exposed even in the presence of a protein corona. As a proof of concept, a weakly basic derivative of DXd (DXdd) is remotely loaded into liposomes, followed by trastuzumab adsorption, achieving 95% encapsulation efficiency for DXdd in 100 nm liposomes (with each trastuzumab molecule delivering approximately 580 DXdd molecules). These trastuzumab-functionalized immunoliposomes exhibit improved tumor inhibition in an SKOV3 tumor model, demonstrating the potential of GA-lipo as a simple and effective approach for constructing targeted nanomedicine delivery systems. This method overcomes key challenges in targeted drug delivery technologies, providing a scalable solution with broad clinical applicability.

Active targeting nanocarriers offer significant potential across various applications by improving the specificity and precision of drugs or nanoparticle delivery to targeted cells or tissues. This enhanced targeting increases the efficiency of intracellular drug delivery through receptor-mediated internalization[1,2]. However, despite substantial research efforts, no active targeting nanocarrier has yet achieved market approval. The main obstacles include the complex design of nanocarriers, which hampers large-scale production, and the formation of a protein corona, a layer of proteins that adsorbs onto the surface of nanoparticles, effectively concealing the targeting molecules[3]. Furthermore, clinical development has been hindered by ineffective patient stratification, where protocols fail to accurately identify patients with the necessary biomarker overexpression to benefit from these therapies[4].

[1]Wuya College of Innovation, Shenyang Pharmaceutical University, Shenyang, China. [2]Department of Oncology, Cancer Stem Cell and Translational Medicine Lab, Innovative Cancer Drug Research and Development Engineering Center of Liaoning Province, Shengjing Hospital of China Medical University, Shenyang, China. [3]Joint International Research Laboratory of Intelligent Drug Delivery Systems, Ministry of Education, Shenyang Pharmaceutical University, Shenyang, China. [4]Key Laboratory of Structure-Based Drugs Design & Discovery of Ministry of Education, Shenyang Pharmaceutical University, Shenyang, China. [5]These authors contributed equally: Jinbo Li, Jiang Yu. ✉e-mail: liuhongzhuo@syphu.edu.cn; liudan@syphu.edu.cn; wangyongjun@syphu.edu.cn

Among the various strategies, antibody-modified nanocarriers have emerged as one of the most promising approaches due to their remarkable selectivity and binding affinity[5–9]. However, conventional methods to prepare these carriers typically involve extensive chemical coupling to attach antibodies to nanoparticle surfaces. While effective in displaying targeting antibodies, these methods face significant industrial challenges, including nanoparticle aggregation, high production costs, and batch-to-batch variability. Additionally, the use of non-natural amino acids during covalent binding can introduce technical difficulties and raise the risk of immunogenicity. Compounding these issues, biomolecular ligand-functionalized nanoparticles often lose their targeting efficiency upon the formation of protein corona, which can mask the surface-bound ligands[10–12].

In contrast, physical adsorption offers a simpler and more flexible alternative for attaching targeting moieties to nanocarrier surfaces[13–16]. Choi et al. demonstrated that adsorbing Herceptin® onto paclitaxel nanocrystals could enhance both the cellular uptake and cytotoxicity of the nanocrystals[13,17]. Du and colleagues developed self-assembled poly (propylene sulfone) nanoparticles capable of supporting the controlled formation of multi-component enzyme and antibody coatings while preserving their bioactivity[16]. Similarly, Tonigold and colleagues synthesized polymeric nanoparticles functionalized with carboxyl or amino surface groups to adsorb targeting antibodies. These findings revealed that pre-adsorbed antibodies not only retained their targeting functionality but also reduced the effects of protein corona formation, compared to antibodies covalently coupled to the nanoparticle surface[18]. However, the broader applicability of these strategies is often restricted by limited in vivo validations[13–19], complex synthesis processes[16], and low payload capacity[16,18,19]. Additionally, physical adsorption is constrained by stability issues, as factors like pH, ionic strength, and surface charge distribution can affect adsorption efficiency. Proteins adsorbed onto nanocarriers are also susceptible to denaturation, which can lead to a loss of bioactivity[20–22].

Therefore, an ideal pre-adsorption targeting platform should combine broad applicability, simple construction, stable adsorption, optimal ligand orientation, and the ability to avoid shielding by the protein corona.

Inspired by the intrinsic binding affinity of polyphenol compounds for proteins[23–26], we synthesize a series of gallic acid-modified lipids (GA-lipids) by screening various types of lipids, including fatty acid lipids and cholesterol (Fig. 1). These GA-lipids are then integrated into lipid materials to create surface-galloylated liposomes (GA-lipo). Pre-adsorption experiments confirm the effectiveness and versatility of GA-lipo in adsorbing proteins. Using doxorubicin (DOX) as the model drug and transferrin as the targeting ligand, we demonstrate the targeting efficacy of this delivery system in preliminary, verified in vivo studies. To further explore the potential of this platform, we develop trastuzumab-based immunoliposomes using GA-lipo as the foundation. GA-lipo is able to stably adsorb trastuzumab while maintaining optimal antibody orientation, crucial for efficient targeting. Furthermore, the GA-lipo system shows resistance to protein corona interference, ensuring that the targeting ligands retain their function and accessibility even after exposure to a biological environment. Overall, this universal, straightforward, and highly efficient strategy holds substantial promise for drug delivery applications and presents a compelling platform for future clinical translation and development.

## Results

### Design and characterization of galloylated liposomes

Our previous studies demonstrated that small-molecule drugs, when modified with gallic acid, exhibited a strong binding affinity to human serum albumin[27]. This finding suggested that galloyl moieties may serve as effective modifiers for adsorbing protein ligands on the nanocarrier surface. To verify this hypothesis, we synthesized a series of lipid galloyl acid derivatives (Fig.2a and Supplementary Figs. 1–8). These derivatives were then incorporated into liposome bilayers,

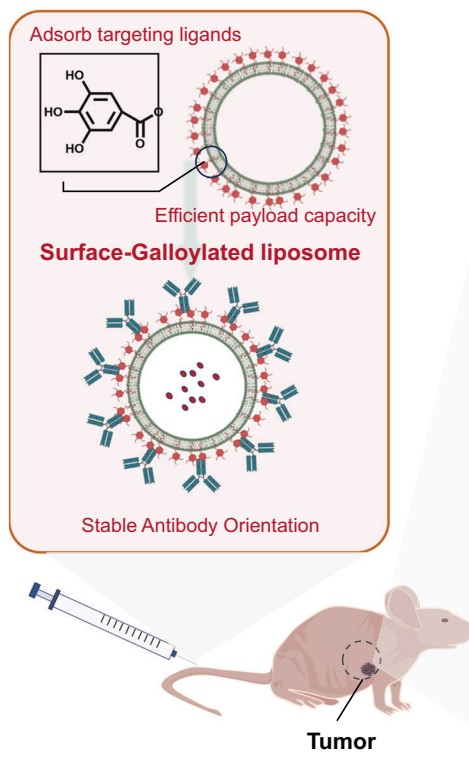

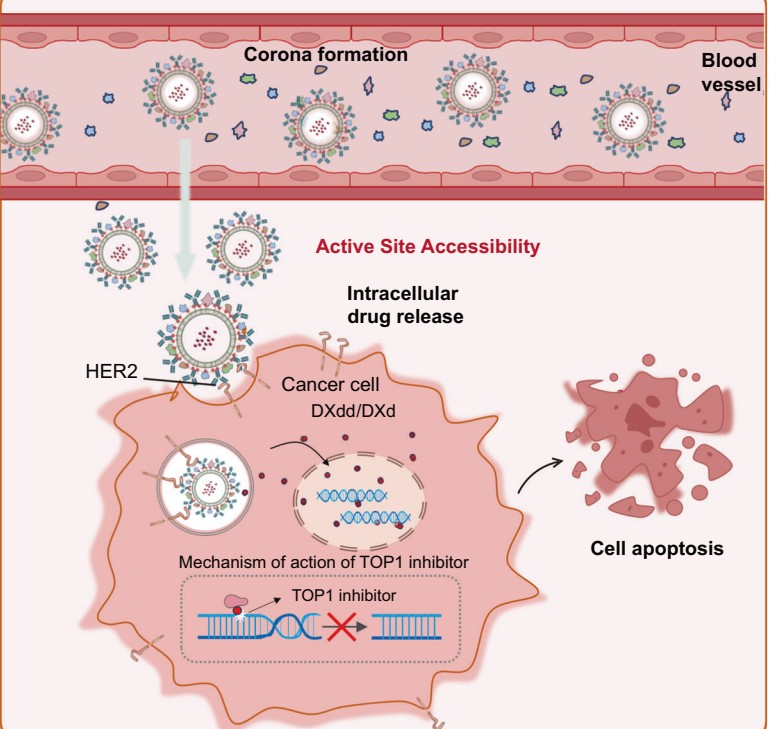

**Fig. 1 | Galloylated liposomes, composed of GA-modified lipids, cholesterol, and HSPC, can pre-adsorb a wide range of proteins.** Specifically, GA-lipo efficiently adsorbs targeting antibodies in a manner that preserves active site accessibility, even following corona formation. Based on this feature, trastuzumab-coated GA-lipo effectively delivered cytotoxic agents to HER2-positive cancer models, resulting in significant tumor inhibition.

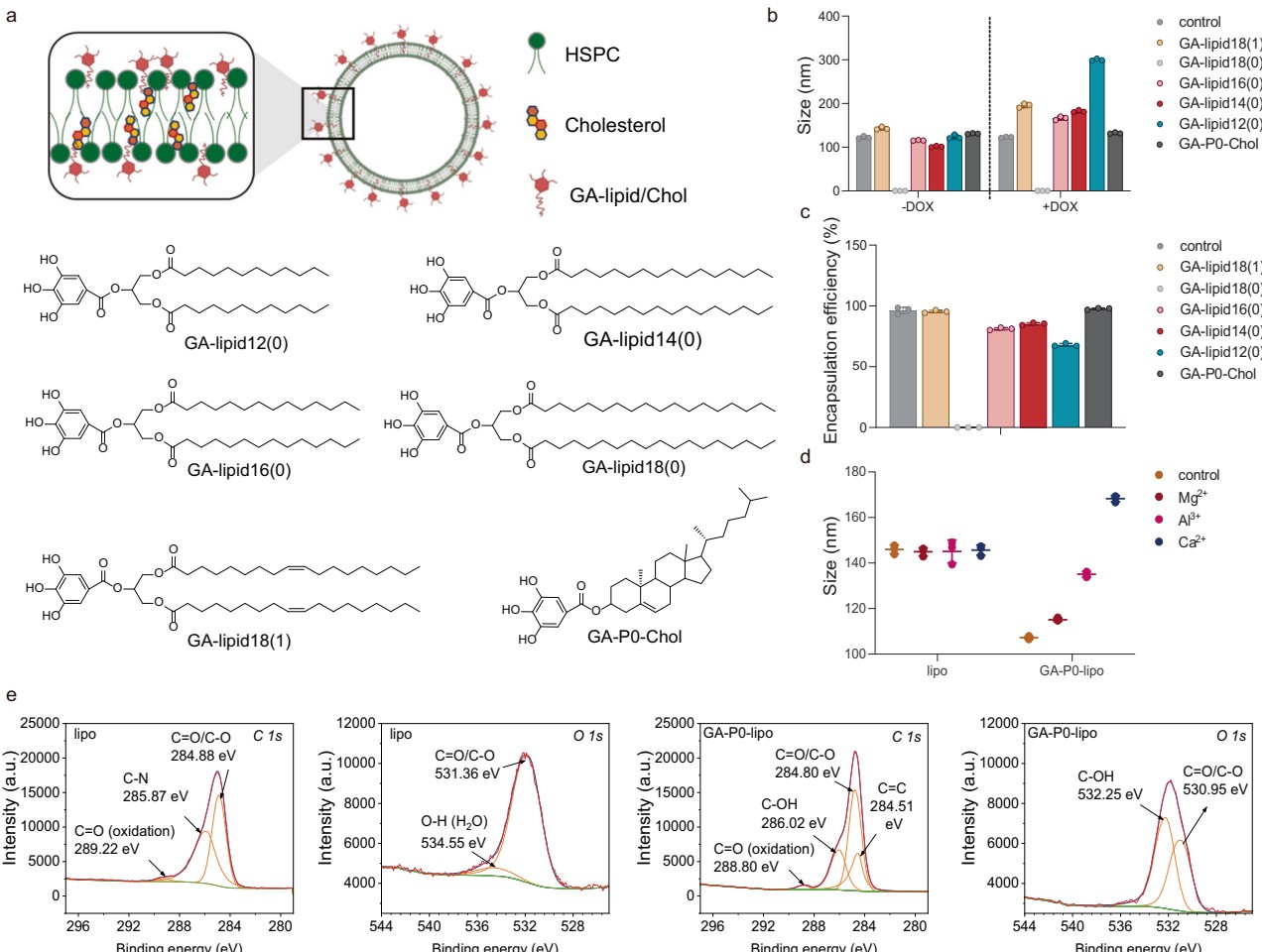

**Fig. 2 | Schematic of protein-coated surface-galloylated liposomes. a** Illustration of surface-galloylated liposomes and the designed structures of lipid galloyl acid derivatives (GA-lipids). **b** Size of blank liposomes and DOX-loaded liposomes (*n* = 3 technical replicates). Data are shown as mean value ± s.d. **c** Encapsulation efficiency of non-galloylated liposome (lipo) and galloylated liposomes (GA-lipo) (*n* = 3 technical replicates). Data are shown as mean value ± s.d. **d** The sizes change of lipo and GA-P0-lipo after incubation with 50 mM metal ions, Mg²⁺, Al³⁺, and Ca²⁺ (*n* = 3 technical replicates). Data are shown as mean value ± s.d. **e** High-resolution individual XPS spectra (*C 1s* and *O 1s*) of lipo and GA-P0-lipo. Source data are provided as a Source Data file.

resulting in the formation of surface-galloylated liposomes. Considering that the hydrophobic segment of GA-lipids might influence the formation and stability of the liposomes, we optimized the GA-lipid structure using several strategies: (i) adjusting the hydrophobic tail length of fatty acid lipids, (ii) introducing unsaturated bonds into the hydrophobic tails of fatty acid lipids, and (iii) screening various fatty acid lipids and cholesterol. These optimizations were essential to ensure the stability and efficacy of the liposomal system, paving the way for efficient protein adsorption and targeted drug delivery.

To identify the optimal GA-lipids and liposome compositions, we evaluated several key parameters, including particle size, stability, and the encapsulation efficiency of model drug doxorubicin (DOX). For GA-fatty acid lipids, both the length and degree of unsaturation in hydrophobic tails significantly influenced liposome stability and DOX encapsulation efficiency (Fig. 2b and Supplementary Table 1). Specifically, increasing the hydrophobic tail length from 12 to 16 carbons resulted in an encapsulation efficiency from 67.04% to 81.26%. However, when the tail length reached 18 carbons, liposome formation was hindered, likely due to the unsuitable critical packing parameter for forming bilayer vesicles. The introduction of unsaturated bonds improved the flexibility of the lipid structure, achieving a maximum encapsulation efficiency of 96.57% for GA-lipo with a molar composition of HSPC, Chol, and GA-lipid18(1) at 60:30:10 (Supplementary Table 2). However, liposomes containing GA-fatty acid lipids exhibited

an increase in particle size from 140 nm to 200 nm after DOX loading (Fig. 2b and Supplementary Table 1), with sedimentation observed after 7 days of storage at 4 °C (Supplementary Fig. 9a). In contrast, GA-cholesterol lipids (GA-P0-Chol) incorporated liposomes showed minimal particle size changes after DOX loading, maintaining a stable size around 130 nm with a loading efficiency of 97.36% (Fig.2b, c, Supplementary Table 3). These results demonstrated the superior stability of GA-lipo incorporating GA-cholesterol lipids (Supplementary Fig. 9b). Based on these findings, we selected GA-cholesterol-modified liposomes for further studies.

UV-vis spectroscopic analysis revealed that the absorption band at 254 nm for GA-lipo was retained, consistent with the GA-lipids (Supplementary Fig. 10). This result indicated that the galloyl acid-modified lipids were successfully incorporated into the phospholipid bilayer. Metal ions can chelate with the surface galloyl moieties, leading to liposome aggregation. The increase in the particle size of GA-liposomes upon the addition of metal ions confirmed the successful exposure of the galloyl moiety on the liposome surface (Fig. 1d). To further characterize the surface properties, X-ray photoelectron spectroscopy (XPS) was employed (Fig. 1e). In comparison with non-galloylated liposomes (lipo), the *C 1s* deconvolutions showed two additional shoulder peaks at 286.02 eV and 284.51 eV, corresponding to C−OH and C=C bonds from the phenolic hydroxyl groups of galloyl units. Furthermore, the presence of phenolic hydroxyl groups on the

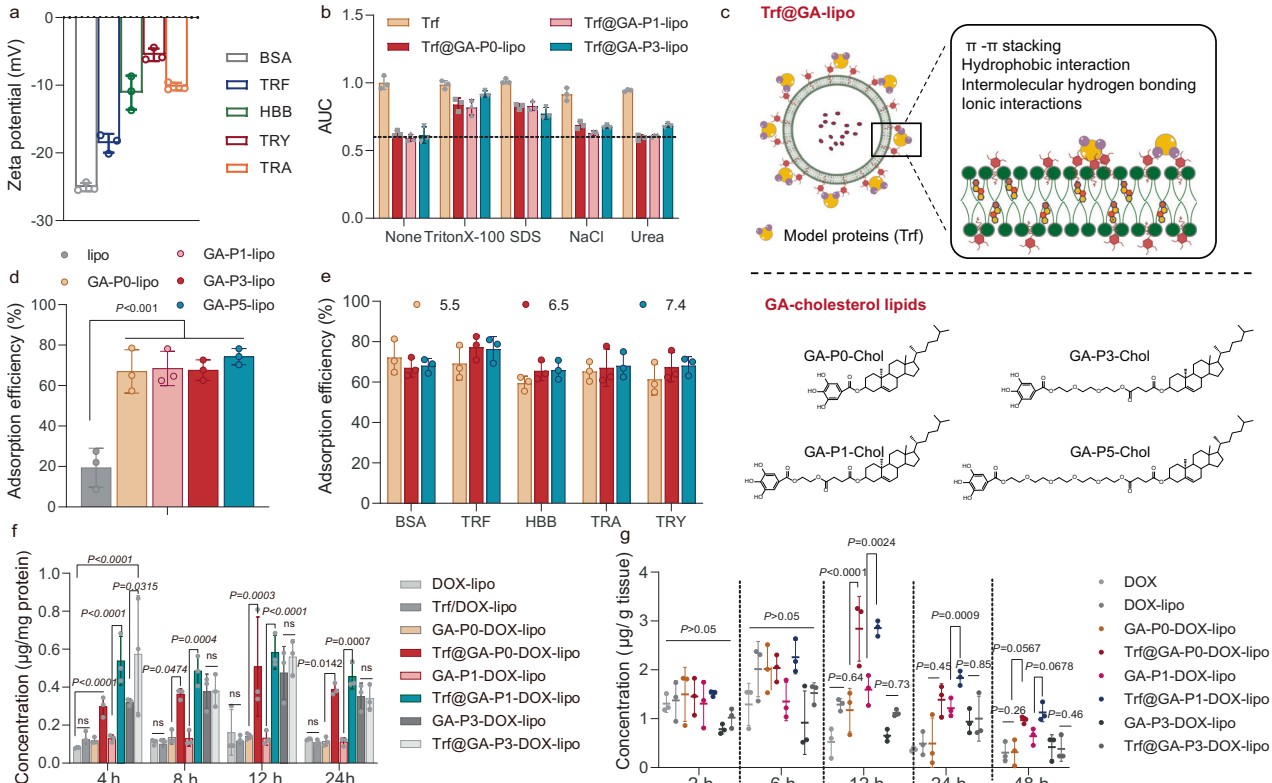

**Fig. 3 | Adsorbed protein on GA-lipo preserves targeting function. a** ζ potential of protein-adsorbed GA-lipo is dependent on the adsorbed protein type. ($n = 3$ technical replicates, mean value ± s.d.). **b** Super centrifugation was used to detect the adsorbed antibodies on the surface of GA-lipo. (pI: Bovine Serum Albumin, BSA 4.7, transferrin, TRF 5.4, Human hemoglobin, HBB, 6.8, Trastuzumab, TRA 8.7, Trypsin 10.1). Data are represented as the MFI of FITC-positive nanoparticles. ($n = 3$ technical replicates, mean value ± s.d.). **c** Structures of galloyl acid-modified cholesterol lipids with different lengths of PEG linker and schematic representation of GA-lipo and their capacity to adsorb proteins. **d** Trf adsorbed onto different GA-Chol modified liposomes. Data are represented as the MFI of FITC-positive nanoparticles. ($n = 3$ technical replicates, mean value ± s.d.). Statistical significance was analyzed by one-way ANOVA and Tukey's multiple comparisons test. **e** Trf Trp quenching recovery after T@GA-lipo incubation with 100 mM Triton X-100, SDS, NaCl and Urea for 4 h. AUC normalized to folded TRA. ($n = 3$ technical replicates, mean value ± s.d.). **f** The result of incubating 4T1 cells with Trf@GA-lipo (5 μg/ml DOX) for 4, 8, 12, 24 h with 10% FBS. The concentration of uptake cells is shown. ($n = 3$ biologically independent samples, mean value ± s.d.). Statistical significance was determined by a two-way analysis of variance with Tukey's multiple comparisons test. **g** The tumor accumulation of Trf@GA-lipo. (5 μg/kg DOX, $n = 3$ biologically independent samples, mean value ± s.d.). Significant differences were determined by two-way ANOVA and Tukey's multiple comparisons test. Source data are provided as a Source Data file.

GA-liposome surface was corroborated by a shoulder peak at 532.25 eV, corresponding to O−H species.

### Establishment of galloylated liposomes for non-covalent protein adsorption

We next explored the capacity of the exposed galloyl moieties on the GA-lipo surface to adsorb proteins non-covalently. A panel of proteins with diverse isoelectric points (pI 4.7–10.1) and molecular weights (23.8–145 kDa) was selected for this investigation. Upon incubation with these proteins, the ζ potentials of GA-lipo varied depending on the specific protein used (Fig. 3a), indicating successful and broad-spectrum protein adsorption. To assess the impact of pH on protein adsorption, we varied the pH of the incubation buffer from 5.5 to 7.4. Importantly, protein adsorption efficiency remained stable between pH 5.5 and 7.4 (Fig. 3b), a physiologically relevant range for drug delivery applications[28]. At pH 6.5–7.4, the selected proteins exhibited different electrical properties: bovine serum albumin (BSA) and transferrin (Trf) were negatively charged, trastuzumab (TRA) and trypsin (TRY) were positively charged, and human hemoglobin (HBB) was electrically neutral. As shown in Fig. 3b, the protein adsorption efficiency of GA-P0-lipo remained largely unchanged within this physiological pH range, which maintained an adsorption efficiency up to 70%, indicating that GA-P0-lipo is capable of adsorbing a broad spectrum of proteins, irrespective of their charge.

To further evaluate the practicality of GA-lipo, we used Trf as a model protein ligand to investigate its targeting potential. Pre-adsorption of Trf onto GA-lipo at 25 °C for 1 h resulted in efficient protein adsorption, forming Trf@GA-lipo (Supplementary Fig. 11a, b). The density of modified GA-Chol was 10.0% (molar ratio), maintaining an adsorption efficiency of approximately 70%. Additionally, a Trf concentration of 0.025% (molar ratio of protein and lipids) demonstrated a superior adsorption efficiency on GA-lipo (Supplementary Fig. 11c, d).

We hypothesized that the distance between the galloyl moiety and cholesterol might influence the galloyl group's exposure on the liposome surface. To test this, we introduced polyethylene glycol (PEG) linkers of varying lengths between the galloyl and cholesterol: no linker (P0), monoethylene glycol (P1), triethylene glycol (P3), and pentaethylene glycol (P5) (Fig. 3c and Supplementary Figs. 12–15). This led to the creation of a series of galloylated liposomes (GA-P-lipo). The successful exposure of galloyl moieties on the surface of all GA-P-lipo variants was confirmed (Supplementary Fig. 16). However, this modification did not result in significant differences in Trf adsorption across the formulations (Fig. 3d and Table S3). When DOX was loaded into these liposomes, no significant differences were observed in either loading efficiency or particle size among formulations with P0, P1, or P3 linkers. All formulations maintained an average size of around 120 nm and achieved encapsulation efficiency of up to 95%. However,

the encapsulation efficiency of DOX-loaded GA-P5-lipo dropped below 90%, reducing to 89.88%. As a result, we selected GA-cholesterol lipids with P0, P1, and P3 variants for further experiment (Supplementary Table 3 and Supplementary Figs. 17, 18).

A tryptophan quenching assay was used to examine whether Trf interacted with the galloyl moieties on GA-lipo. Tryptophan fluorescence quenching of Trf upon adsorption to GA-lipo confirmed direct interaction (Supplementary Fig. 19a). Addition of agents disrupting non-covalent interactions partially restored fluorescence, implicating hydrophobic interactions as the dominant adsorption force, along with hydrogen bonding and electrostatic interactions (Fig. 3e). FTIR spectra of Trf@GA-lipo exhibited clear shifts in the characteristic amide I (-1640 cm$^{-1}$) and amide II (-1543 cm$^{-1}$) bands, indicating specific interactions between the protein backbone ($-NH_2$ groups) and the polyphenol moieties on the GA-lipo surface (Supplementary Fig. 19c). CD spectra of Trf@GA-lipo remained nearly identical to that of the free protein, indicating preservation of its secondary structure upon adsorption (Supplementary Fig. 19d).

In cell uptake studies, Trf@GA-P0-lipo and Trf@GA-P1-lipo significantly enhanced DOX internalization relative to non-protein-modified GA-lipo (Fig. 3f). However, no significant difference in DOX uptake was observed between Trf@GA-P3-lipo and GA-P3-lipo, which may be attributed to the instability of GA-P3-lipo, as indicated by a notable increase in particle size during storage at 37 °C (Supplementary Fig. 20). The reduced stability associated with longer PEG linker (greater than one unit) may be due to hydrogen bonding between the galloyl moiety and PEG, disrupting the integrity of phospholipid bilayer[29,30]. In contrast, the P1 linker is too short to form such interactions, resulting in comparable stability to P0 formulations. Altogether, for optimal liposome stability, the hydrophilic PEG linker between the galloyl and cholesterol should not exceed a length of one unit.

To investigate whether the adsorbed Trf would be competitively displaced by serum proteins, we incubated Trf@GA-lipo with plasma at 37 °C. A qualitative analysis of the adsorbed proteins on the liposome surface was conducted using SDS-PAGE. SDS-PAGE revealed Trf retention on the liposome surface after protein corona formation (Supplementary Fig. 21). Additionally, we adsorbed GA-lipo with fluorescein isothiocyanate (FITC)-labeled Trf and incubated these FITC-Trf@GA-lipo with plasma at 37 °C. Measurements of FITC-Trf in the supernatants showed no significant displacement after 48 h of incubation, indicating that Trf@GA-lipo remained stable under physiological conditions and was less affected by protein corona formation (Supplementary Fig. 22).

To validate these findings in vivo, we assessed the therapeutic efficacy of Trf@GA-DOX-lipos in a 4T1 subcutaneous tumor model. Female BALB/c mice bearing 4T1 tumors (-100 mm$^3$) were intravenously administered liposomal formulations at a DOX-equivalent dose of 5 mg/kg every three days for a total of four injections. No significant differences in tumor volume were observed between the DOX-lipo, Trf/DOX-lipo, GA-DOX-lipo (P3), and Trf@GA-DOX-lipo (P3) groups (Supplementary Fig. 23a). In contrast, the tumor volumes of the Trf@GA-DOX-lipo (P0) and (P1) groups were 1.4- and 2.5-fold smaller, respectively, compared to the untargeted GA-DOX-lipo groups, indicating enhanced therapeutic efficacy with good safety (Supplementary Fig. 23b). Further analysis of pharmacokinetics, drug distribution and tumor accumulation (Fig. 3g, Supplementary Fig. 23c, Fig. 24 and Table 4) showed that Trf@GA-lipo (P0, P1) exhibited a greater area under the curve (AUC), higher tumor accumulation and reduced major organ accumulation than other groups, further supporting the improved performance of the targeted delivery system.

## Surface-galloylated liposomes enable stable orientation of pre-adsorbed antibodies

To investigate the feasibility of adsorbing an antibody onto GA-lipo, we tested the efficacy of monoclonal antibody (mAb) adsorption using trastuzumab (TRA) as a model. GA-P1-lipo was selected as the optimal liposomal formulation for mAb adsorption (T@GA-P1-lipo) (Supplementary Fig. 25a). Pre-incubation of GA-P1-lipo with TRA at 25 °C for 1 h resulted in efficient adsorption (>70%) at an initial TRA concentration of 0.05 mol%, which is sufficient for targeted drug delivery[31] (Supplementary Fig. 25b). Based on these findings, we developed a flexible and versatile platform for mAbs adsorption onto GA-lipo, enabling efficient targeted drug delivery. In this process, drug-loaded GA-lipo was incubated with mAbs at room temperature for 1 h, followed by a washing step to remove residual proteins. The successful presence of TRA on GA-lipo surface was confirmed through staining with secondary gold-coupled antibodies (Fig. 4a). Trastuzumab adsorption caused a shift in ζ potential from −20.9 mV to −9.52 mV (Fig. 4b). Additionally, the quenching of tryptophan fluorescence in TRA after adsorption onto GA-lipo indicated strong interactions between GA and TRA (Supplementary Fig. 26a). T@GA-lipo displayed similar FTIR band shifts, reflecting analogous molecular interactions, yet the CD spectra of the adsorbed trastuzumab remained nearly identical to that of the free antibody, indicating preservation of its secondary structure upon adsorption (Supplementary Fig. 26c, d).

The IgG molecule possesses a Y-shaped structure, where the antigen-binding domain (Fab) specifically recognizes tumor-associated antigens and modulates downstream signaling pathways. The crystallizable fragment (Fc), a conserved region, binds to immune cells expressing Fc receptors and to complement proteins in the blood[32]. We hoped that the Fab region of adsorbed IgG on GA-lipo would be exposed outward, a critical factor for enhancing liposomes' internalization into tumor cells (Fig. 4c). To test this, we used flow cytometry to assess the orientation of TRA adsorbed on GA-lipo by labeling with specific anti-Fab and anti-Fc antibodies. Minimal secondary antibody labeling (<1%) was detected in non-galloylated liposomes (lipo) and TRA/lipo mixtures, whereas 15% phycoerythrin (PE)-labeled liposomes were observed in GA-P1-lipo. In contrast, approximately 60% of T@GA-P1-lipo were stained with anti-Fab antibodies, similar to the staining observed with directly FITC-labeled TRA before incubation (Fig. 4d, e). Consistent results were obtained with anti-Fc antibodies, with negligible binding of anti-Fc antibody detected in any of the liposome groups (Fig. 4f). These findings suggest that the Fc region of IgG1 primarily complexes with galloyl moieties on GA-P1-lipo. The preference of the Fc region for galloyl binding may be attributed to its higher hydrophobicity compared to the Fab region in IgG1 molecules[33], aligning with the understanding that hydrophobic interactions are the primary force driving protein adsorption. To further investigate the interaction types and binding sites between the gallic acid moiety and trastuzumab, we performed molecular dynamics (MD) simulations. The results showed that both Coulombic (electrostatic) and Lennard-Jones (van der Waals, including hydrophobic) interactions stabilize the complex, while hydrogen-bond counts steadily rise during the trajectory. π−π stacking is negligible because GA-P1-Chol contains a single phenyl ring, yet its contribution is implicitly captured in the van-der-Waals term (Fig. 4g–j, Supplementary Figs. 27 and 28). The stable binding configurations of Trastuzumab with GA-cholesterol are presented in Fig. 4g. The Fc domains (domains 5–8) display markedly lower total interaction energies than either Fab arm, indicating higher affinity (Fig. 3i). Per-domain contact maps (Supplementary Fig. 28b, c) reveal dense hydrophobic pockets together with recurrent H-bonds (e.g., D633, E434, T301) unique to Fc, rationalizing the selectivity. Collectively, these quantitative MD data corroborate our initial hypothesis: hydrophobic contacts constitute the dominant driving force, complemented by hydrogen bonding and electrostatic interactions, yielding a net preference for the Fc region.

A filler/blocking protein is usually used to cover locations that are not bound by the desired antibodies[16]. We investigated whether blocking unbound surface sites with proteins like BSA is necessary for

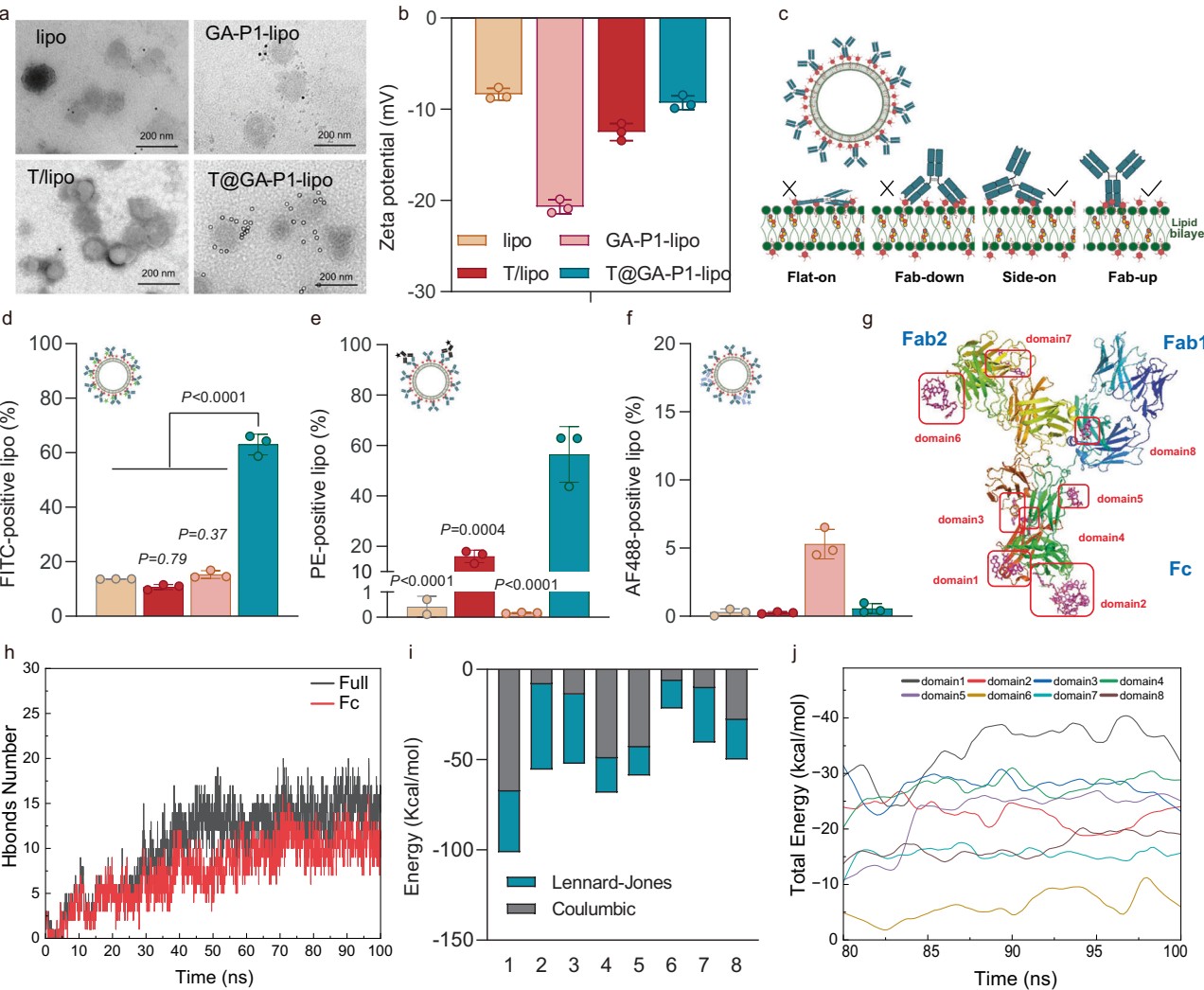

**Fig. 4 | Orientation of pre-adsorbed antibodies on the GA-lipo. a** Immunogold labeling showing the binding of pre-adsorbed TRA antibodies to secondary antibodies. **b** ζ potential of liposomes and TRA adsorption liposomes (*n* = 3 technical replicates, mean value ± s.d.). **c** Illustration of the orientation of pre-adsorbed antibodies on surface-galloylated liposomes. **d** Detection of TRA with fluorescently (FITC) labeled by flow cytometry. Data are shown as the amount of FITC-positive liposomes (*n* = 3 technical replicates, mean value ± s.d.). **e** Labeled adsorbed TRA with secondary-anti-human IgG1 targeting Fab of TRA measured by flow cytometry. Data are shown as the amount of PE-positive liposomes (*n* = 3 technical replicates, mean value ± s.d.). **f** AF488-labeled secondary-anti-human IgG1 targeting Fc of TRA. Data are shown as the amount of PE-positive liposomes (*n* = 3 technical replicates, mean value ± s.d.). Statistical significance (**d**–**f**) was determined by a two-way analysis of variance with Tukey's multiple comparisons test. **g** Simulation snapshot of

an equilibrated TRA protein bound to twenty GA-P1-Chol molecules (in purple). The ligand–protein interfaces are stabilized by hydrophobic pockets and hydrogen-bond networks, as indicated from domain 1 to domain 8, maintaining the structural integrity of TRA. **h** Time evolution of hydrogen bonds formed between GA-P1-Chol and the full-length trastuzumab (black) or its Fc region (red) during a 100 ns MD simulation. The increasing trend indicates progressive stabilization of the complex over time, with a higher number of hydrogen bonds observed in the full antibody structure. **i** Decomposition of binding energy contributions between GA-P1-Chol and each domain (1–8), showing Coulombic (electrostatic) and Lennard-Jones (van der Waals) interactions. **j** Time-resolved total interaction energy (kcal/mol) between GA-P1-Chol and individual antibody domains during the last 20 ns of MD simulation. Source data are provided as a Source Data file.

T@GA-lipo. Testing this approach revealed that BSA blocking had no significant impact on cellular uptake, indicating it is not required in our system (Supplementary Fig. 30a). Although serum proteins may adsorb to unblocked surfaces, their presence does not appear to interfere with the targeting function. To assess whether Fab-outward orientation enhanced targeting efficiency, confocal microscopy showed that both the DiR-labeled liposomes and FITC-labeled TRA in T@GA-P1-lipo co-localized with lysosomes after 8 h of co-incubation in HER2-positive SKOV3 cells, indicating effective internalization (Supplementary Fig. 29a). In contrast, no significant difference in DiR uptake was observed between T@GA-P1-lipo and other groups in HER2-negative MCF-7 cells (Supplementary Fig. 29b), confirming that TRA adsorbed on GA-lipo facilitates the endocytosis of T@GA-P1-lipo specifically in HER2-positive cells. Furthermore, to confirm specific

targeting, we performed competitive inhibition experiments in HER2-positive SKOV3 cells. Cells were pre-treated with excess anti-HER2 antibody to block receptor binding prior to incubation with T@GA-lipo. Flow cytometry analysis showed significantly reduced cellular uptake under HER2 blockade (Supplementary Fig. 30), confirming that internalization is driven by antibody–antigen interactions rather than passive diffusion. To assess non-specific Fc receptor-mediated uptake, THP-1-derived macrophages were used to evaluate both liposome internalization and cytokine secretion[34]. Flow cytometry analysis showed no increase in T@GA-lipo uptake compared to non-targeted liposomes, indicating minimal FcγR interaction (Supplementary Fig. 31a). In parallel, cytokine profiling by ELISA revealed no significant elevation of TNF-α or IL-6, in contrast to the trastuzumab and T/lipo groups containing unabsorbed antibody (Supplementary Fig. 31b, c).

IL-10 levels also remained unchanged, suggesting no detectable immunostimulatory effects.

Then, the targeted delivery performance of GA-lipo with conventional covalent conjugation liposomes was compared. We systematically evaluated the physicochemical stability, cellular uptake and in vivo biodistribution of T@GA-lipo versus T@Mal-lipo (Supplementary Fig. 32). The results demonstrated that both T@GA-lipo and T@Mal-lipo maintained comparable hydrodynamic size (120–160 nm), low PDI (<0.2) and slightly negative ζ-potential (−14 to −5 mV) over 48 h, indicating similar colloidal stability (Supplementary Fig. 32a). In HER2-overexpressing SKOV3 cells, T@GA-lipo achieved higher DiO-positive cell percentages than T@Mal-lipo at all serum levels tested. The difference became most pronounced in 50% FBS, where uptake of T@GA-lipo remained >70%, whereas T@Mal-lipo dropped to <20% (Supplementary Fig. 32b, c). Live imaging of DiR-labeled formulations in SKOV3 xenograft mice showed that both groups accumulated rapidly in tumors, reaching maximum radiance between 4 and 8 h. Importantly, the signal of T@GA-lipo persisted through 36 h and 48 h, whereas that of T@Mal-lipo declined significantly ($p = 0.0414$ and 0.0006 at 36 h and 48 h, respectively; Supplementary Fig. 32d, e). Ex vivo quantification at 48 h revealed notably lower liver and spleen fluorescence for T@GA-lipo ($p < 0.0001$), with no statistical difference in heart, lung, or kidney, confirming reduced RES uptake and enhanced tumor selectivity (Supplementary Fig. 32f). These data highlight the superior serum-tolerance of the GA-lipo interface. Taken together, these results demonstrate that GA-lipo matches the physical stability of covalent conjugates while offering superior serum-resistant cellular uptake, prolonged tumor residence, and diminished off-target accumulation.

## Adsorption of monoclonal antibody ensures active site accessibility even after protein corona formation

Inspired by Enhertu, a clinically approved ADC, we selected DXd as our cytotoxic agent. Building on our previous research, we designed a weakly basic derivative of DXd, termed DXdd, to facilitate its remote loading into liposomes[35–38] (Supplementary Figs. 33 and 34). Using sucrose octasulfate octatriethylamine salt (SOS-TEA) as the inner phase, DXdd was successfully incorporated into liposomes, achieving an encapsulation efficiency of up to 95% at a DXdd-to-lipid ratio of 1:10 (≈14,000 DXdd molecules per liposome) (Table S5). No significant differences in loading efficiency or drug release profiles were observed between DXdd-loaded liposomes (DX-lipo) and their galloylated counterparts (GA-DX-lipo) (Table S5 and Supplementary Fig. 35a). The physicochemical stability of GA-DX-lipo and DX-lipo was monitored over a period of 90 days. Dynamic light scattering (DLS) measurements showed no significant changes in particle size, PDI, or zeta potential at either 4 °C or 25 °C (Supplementary Fig. 35b, c), indicating good colloidal stability during storage. Furthermore, Cryo-TEM analysis revealed that GA-DX-lipo retained intact vesicular morphology after 3 months at 25 °C, with no detectable difference from freshly prepared liposomes (Supplementary Fig. 35d). These results suggest that the formulation maintains good physical integrity over an extended period and supports its potential for further shelf-life development. Colloidal gold labeling confirmed efficient post-loading adsorption of trastuzumab (TRA), with ~70% of TRA (equivalent to ~49 molecules per liposome) successfully immobilized at an initial antibody concentration of 0.05 mol% (Table S5 and Supplementary Fig. 36c). The resulting liposomes had an average diameter of approximately 100 nm (Supplementary Fig. 36a–e). Cellular uptake studies revealed that an efficient initial incubating TRA concentration for targeting was 0.025 mol% (≈24 antibodies per liposome), suggesting that each antibody on GA-lipo could efficiently carry approximately 580 drug molecules for targeting delivery (Supplementary Fig. 37).

To investigate whether protein corona formation affected the uptake of T@GA-DX-lipo, we examined the influence of fetal bovine serum (FBS) concentrations in the culture medium, ranging from 0% to 50% (Fig. 5a–c). Notably, trastuzumab-adsorbed liposomes exhibited more than 2-fold increase in DXdd uptake across different FBS concentrations compared to other groups, demonstrating the robust targeting capability of our formulation despite the formation of protein corona.

It is well known that the formation of biomolecular corona on the surface of nanoparticles can mask their target protein, thereby reducing targeting efficiency[12]. So, why does our mAb-adsorbed GA-lipo maintain its targeting function despite the presence of biomolecular corona? The answer likely lies in the oriented arrangement of the pre-adsorbed mAbs on the GA-lipo surface[18]. Characterization of the hard protein corona revealed that the adsorbed TRA on the GA-lipo surface was not fully replaced by serum proteins (Fig. 5d and Supplementary Fig. 38). To further assess the binding stability of TRA, isothermal titration calorimetry (ITC) analysis was performed, yielding a dissociation constant (KD) of 0.329 μM, indicative of strong affinity between TRA and GA-lipo (Supplementary Fig. 39a, b and Table 6). In addition, SDS-PAGE-based time-course profiling of the protein corona (from 5 min to 24 h) confirmed that while the total protein composition on the liposome surface evolved over time, the intensity of the trastuzumab band remained largely unchanged (Supplementary Fig. 39c). This result indicated that trastuzumab remains stably adsorbed on GA-lipo despite the presence of competing serum proteins. In fact, over 65% of TRA was retained on the GA-DX-lipo after incubation with serum (Supplementary Fig. 40). Importantly, more than 30% T@GA-lipo were stained with anti-Fab antibodies, indicating that the active Fab region of TRA adsorbed on GA-lipo remained exposed, even in the presence of the protein corona (Fig. 5e). Additionally, the antibody-modified formulation (T@GA-DX-lipo) showed even greater size stability, likely due to enhanced interfacial stabilization conferred by the galloyl-antibody interactions. (Supplementary Fig. 41).

We further investigated the adsorption behavior of GA-lipo using another monoclonal antibody, cetuximab (Cet) (Supplementary Fig. 42). Similar results were observed, showing that cetuximab adsorbed GA-P1-lipo (C@GA-P1-lipo) could tolerance the formation of protein corona while still maintaining the exposure of the Fab regions of the pre-adsorbed antibodies (Fig. 5f). The orientation of Cet was also found to involve an interaction between the galloyl moiety and the Fc region of the antibody (Fig. 5g, h). This stable orientation of pre-adsorbed antibodies ensures that the active Fab sites remain exposed, allowing for effective targeted delivery (Fig. 5i and Supplementary Fig. 43).

To validate the generalizability of our GA-lipo platform across antibodies with different physicochemical properties, we further examined two additional therapeutic monoclonal antibodies—nimotuzumab (pI ≈ 7.5) and rituximab (pI ≈ 9.1)—in addition to trastuzumab (pI ≈ 8.8) and cetuximab (pI ≈ 8.7). After 1 h incubation at 25 °C, GA-P1-lipo exhibited high adsorption efficiency for nimotuzumab (64.33 ± 6.03%) and rituximab (64.56 ± 4.35%) (Supplementary Table 7, Supplementary Fig. 44). These values are comparable to those of trastuzumab (69.82 ± 7.18%) and cetuximab (66.87 ± 8.14%), confirming that the platform accommodates antibodies with diverse pI values and structural features. As shown in Supplementary Fig. 45, all four antibody-modified liposomes (N@, R@, T@, C@GA-P1-lipo) maintained stable particle size (generally within 150–180 nm), low PDI (<0.25), and moderate zeta potential (−14 to −5 mV) over 48 h in PBS containing 10% FBS at 37 °C, indicating good colloidal stability under physiological conditions. The fluorescence signal of FITC-labeled nimotuzumab and rituximab on GA-lipo remained largely intact after 2 h incubation with 50% mouse plasma, indicating strong binding and minimal displacement by serum proteins (Supplementary Fig. 46b, d). As summarized in Supplementary Table 7, three independently prepared batches of each mAb@GA-P1-lipo formulation showed

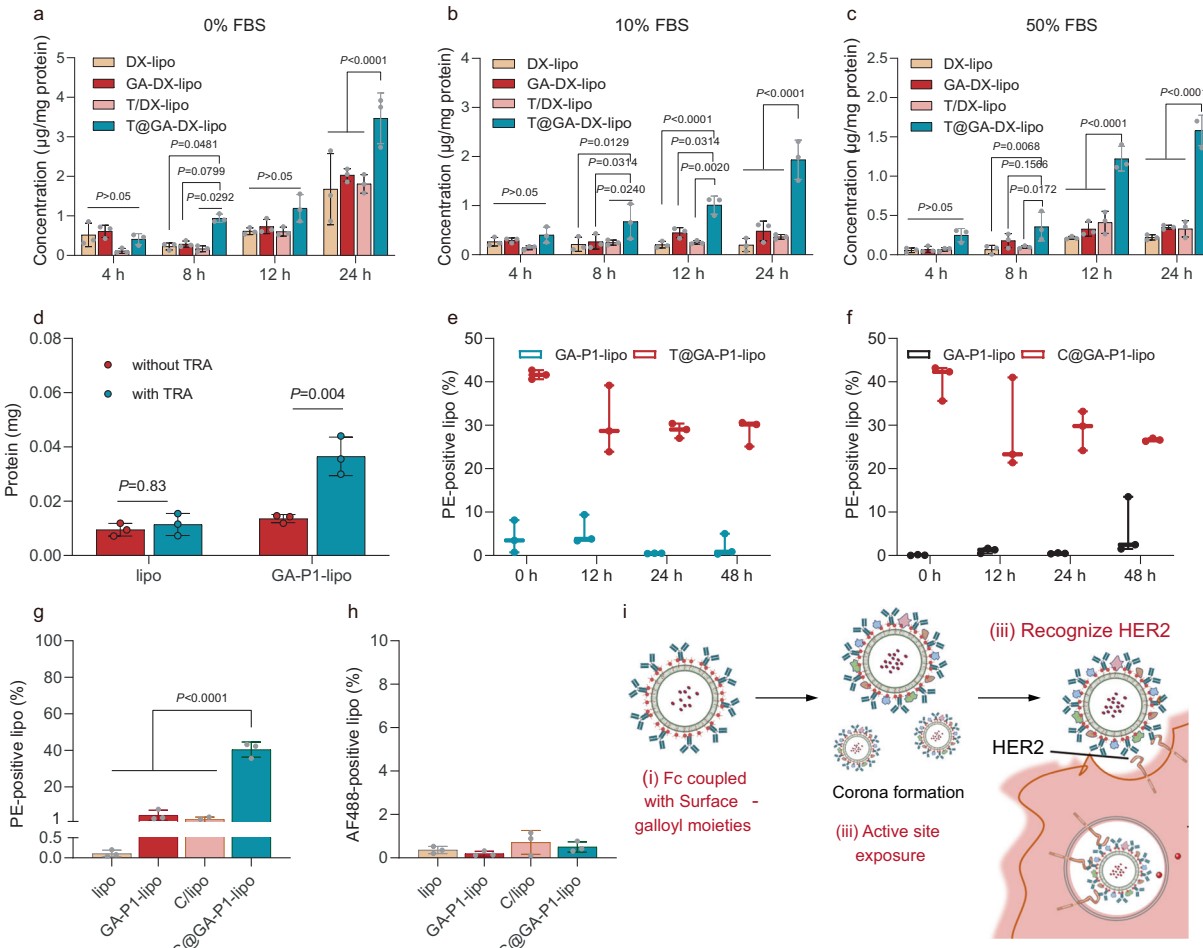

**Fig. 5 | The coated antibody on the GA-lipo retained the targeting bioactivity.**
**a**–**c** Incubation of SKOV3 cells with DX-loaded liposomes (DXdd 5 µg/ml) for 4, 8, 12, 24 h, 37 °C in the presence of different FBS concentrations. **a** 0% FBS, **b** 10% FBS, **c** 50% FBS ($n = 3$ biologically independent samples, mean value ± s.d.). Statistical significance was analyzed by one-way ANOVA and Tukey's multiple comparisons test. **d** The amount of Hard protein corona of various liposomes. The amount of protein is measured with BCA kits and showed as mean ± s.d. ($n = 3$ technical replicates). Statistical significance was analyzed by one-way ANOVA and Tukey's multiple comparisons test. **e**, **f** the exposure of Fab of mAb on the GA-lipo surface in 50% FBS at 37 °C. labeled coated mAb with secondary-anti-human IgG1 targeting Fab of mAb measured by flow cytometry. Data are shown as the amount of PE-positive liposomes ($n = 3$ technical replicates, mean value ± s.d.). **e** Trastuzumab,

**f** Cetuximab. **g** labeled coated cetuximab with secondary-anti-human IgG1 targeting Fab of cetuximab measured by flow cytometry. Data are shown as the amount of PE-positive liposomes ($n = 3$ technical replicates, mean value ± s.d.). Statistical significance was analyzed by one-way ANOVA and Tukey's multiple comparisons test. **h** AF488-labeled secondary-anti-human IgG1 targeting Fc of cetuximab. Data are shown as the amount of AF488-positive liposomes ($n = 3$ technical replicates, mean value ± s.d.). Statistical significance was determined by a two-way analysis of variance with Tukey's multiple comparisons test. **i** Illustration of the mechanism of overcoming the shielding of protein corona by pre-adsorption Antibody. The orientation of pre-adsorbed antibodies is stable. Even with corona formation, the active site can be exposed outside the liposomes, which makes the successful recognition of the targeted receptor. Source data are provided as a Source Data file.

consistent particle size (CV% all <4%) and adsorption efficiency (CV% ranging from 5.8% to 12.1%). These results confirm the reproducibility and robustness of the platform for antibody surface adsorption.

### Pre-adsorption antibody of surface-galloylated liposome for cancer therapy

Building on the success of our strategy in overcoming the shielding effects of the protein corona on T@GA-DX-lipo, we evaluated the therapeutic efficacy of this approach in a tumor model. First, we assessed the cytotoxicity of DXdd-loaded liposomes in cell lines with varying levels of HER2 expression. In Fig. 5a, the cytotoxicity of GA-DX-lipo appeared to be higher than that of DX-lipo, despite similar release behaviors between the two formulations. The reason may be attributed that the gallic moiety acts as a "recognition enhancer" that amplifies tumor cell-selective uptake via physicochemical interactions (H-bonding/vdWF) and biological targeting (EGFR/ECM affinity)[25,39]. This leads to increased intracellular delivery of DXdd, thereby

enhancing cytotoxicity (Supplementary Fig. 47). For the targeted liposome, T@GA-DX-lipo exhibited the highest cytotoxicity in SKOV3 cells, outperforming GA-DX-lipo, DX-lipo, and T/DX-lipo. In contrast, in MCF-7 cells, there were no significant differences in the IC50 values between targeted and untargeted liposomes (Fig. 6a). These results confirm the efficacy of our targeting strategy in vitro.

Next, we examined the biodistribution of DiR-labeled liposomes using in vivo imaging in BALB/c nude mice bearing SKOV3 xenograft tumors. Mice were intravenously injected with DiR-labeled liposomes (DiR: 1 mg/kg). T@GA-lipo achieved maximal tumor accumulation within 4 h post-injection, while other liposomal formulations peaked at 24 h. Throughout the study, T@GA-lipo consistently demonstrated higher tumor accumulation compared to other formulations, highlighting its superior targeting capability (Fig. 6b, c). At 48 h post-injection, imaging of major organs revealed reduced liver accumulation of T@GA-lipo compared to other groups, suggesting enhanced tumor targeting (Fig. 6b, d). To further evaluate organ-specific

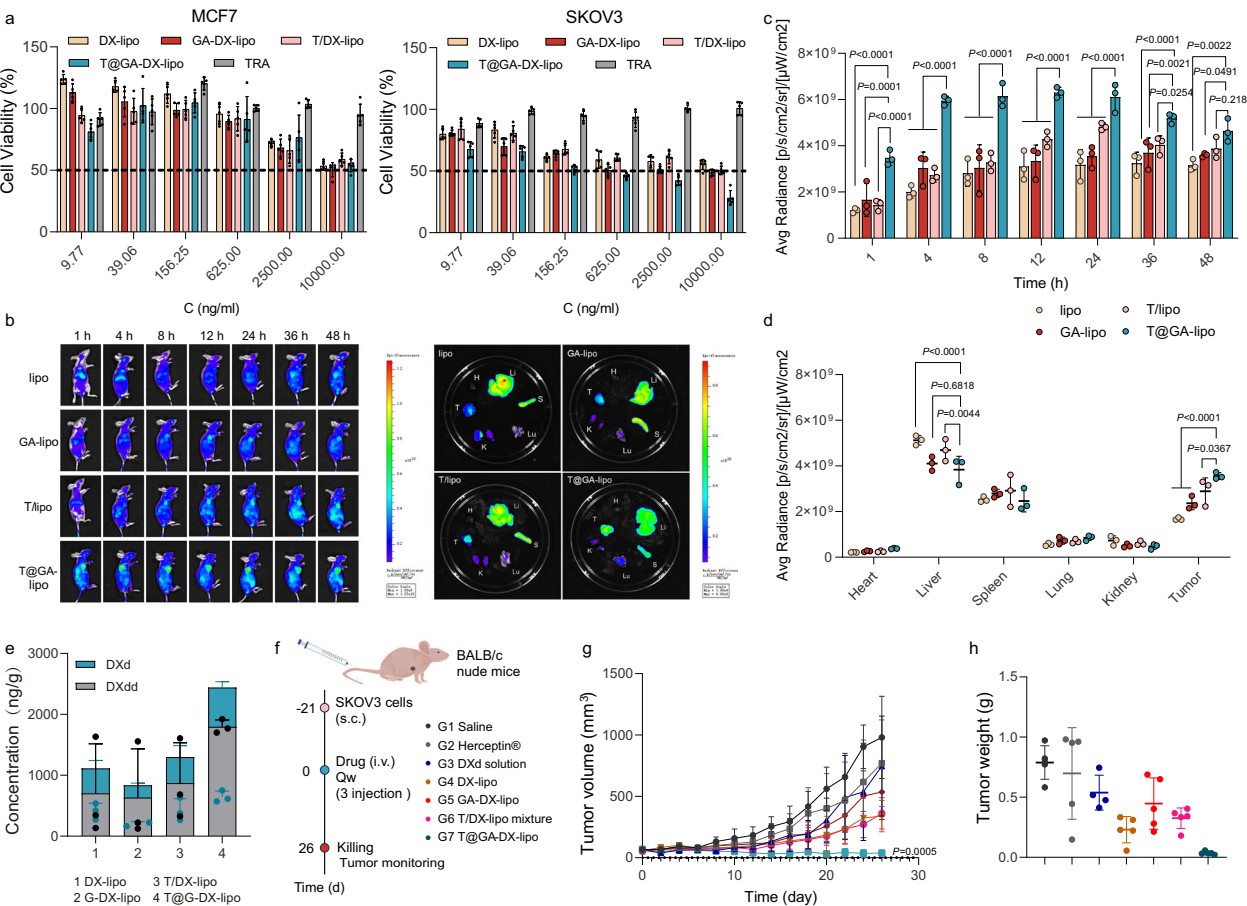

**Fig. 6 | Treatment efficacy of TRA adsorbed galloylated liposomes in SKOV3 tumor-bearing nude mice. a** Cell viability treated with various concentrations of targeted and untargeted DX-lipo and GA-DX-lipo in MCF-7 and SKOV3 cells after 48 h. ($n$ = 5 technical replicates, mean value ± s.d.). **b** In vivo fluorescence imaging of DiR-labeled liposomes in the tumor at different determined times after injection administration (1 mg/kg DiR) and Ex vivo fluorescence imaging of major organs obtained from the mice at 48 h post-injection. Three biological replicates were measured ($n$ = 3 biologically independent samples). **c** Semi-quantitative analysis of the fluorescent intensity of DiR-labeled liposomes in the tumor at different times after injection administration by in vivo fluorescence imaging. ($n$ = 3 biologically independent samples). **d** Semi-quantitative analysis of the fluorescent intensity of major organs obtained from the mice at 48 h post-injection via Ex vivo fluorescence imaging. ($n$ = 3 biologically independent samples, mean value ± s.d.). Statistical

significance (**c**) and (**d**) was analyzed by two-way ANOVA and Tukey's multiple comparisons test. **e** Biodistribution of DXdd and DXd in the tumor at 24 h post-injection. Three biological replicates were measured ($n$ = 3, mean value ± s.d.; one-way ANOVA). **f** Scheme and grouping of in vivo therapy. The BALB/c nude mice were inoculated subcutaneously into the right axillary region (underarm area) with SKOV3 cells ($5 \times 10^6$ cells per mouse) on day −21, and the mice were treated with different preparations (a DXdd concentration of 5 mg/kg, TRA, 5 mg/kg) Qw over the course of 21 days. **g** Tumor volume of SKOV3 tumor in mice after drug administration. Statistical significance was analyzed by two-way ANOVA and Tukey's multiple comparisons test. **h** Tumor weight of tumors obtained after treatment cycles in respective groups. (n = 5 biologically independent samples, mean value ± s.d.; one-way ANOVA). Source data are provided as a Source Data file.

distribution and clearance, we conducted biodistribution studies of T@GA-DX-lipo by quantifying both the derivative (DXdd) and the active drug (DXd) in major organs (Supplementary Fig. 48). Compared to non-targeted formulations, targeted liposomes markedly reduced drug accumulation in off-target organs, such as the liver and spleen, while enhancing drug retention in tumor tissues (Fig. 6e and Supplementary Figs. 48). Notably, drug concentrations in most organs peaked between 12 and 24 h post-injection, followed by a consistent decline at 48 h, indicating progressive clearance. These findings highlight the improved biodistribution profiles of the targeted liposomes, supporting the potential to minimize systemic toxicity and enhance tumor-selective delivery of the pre-adsorption targeting strategy.

Finally, we evaluated the therapeutic efficacy of this approach in SKOV3 tumor-bearing nude mice (Fig. 6f). The 5 mg/kg dose was selected on the basis of an MTD (maximum tolerated dose) study performed in female BALB/c-nu mice. Free DXd at 10 mg/kg caused acute toxicity and mortality, whereas DXdd-loaded liposomes were

non-lethal up to the same dose but produced significant weight loss. Doses of 3 mg/kg and 5 mg/kg were well tolerated, showing no behavioral changes or body-weight reduction (Supplementary Fig. 49). We therefore chose 5 mg/kg as an efficacy dose that balances potency and safety. Trastuzumab (Herceptin®) and DXd solutions exhibited minimal therapeutic effects, with one mouse in the DXd group succumbing to severe toxicity after the first treatment. The tumor volumes in the DX-lipo, GA-DX-lipo, and T/DX-lipo groups were reduced by 2.8-, 1.8-, and 2.7-fold, respectively, compared to the saline control group. Notably, tumors in the T@GA-DX-lipo group did not grow during the treatment cycle, resulting in an up to 8-fold reduction in size compared to the saline and other treatment groups (Fig. 6g and Supplementary Fig. 50). Additionally, the tumor weights in the T@GA-DX-lipo group were below 0.03 g (Fig. 6h).

Histological analysis revealed significant tumor tissue damage and a reduction in the Ki67 proliferation index in the T@GA-DX-lipo group compared to the other groups (Supplementary Fig. 51). The levels of HER2 expression in the T@GA-DX-lipo group were comparable to

those of other groups, confirming the effectiveness of HER2 targeting (Supplementary Fig. 51). Importantly, no significant differences in body weight were observed across the groups, and no major organ damage was detected (Supplementary Figs. 52, 53). Normal hepatic and renal function, along with minimal hematological toxicity, further supported the in vivo safety profile of the DXdd-loaded liposomes (Supplementary Figs. 54 and 55). We investigated the cytokine level in serum samples from treated mice. As shown in Supplementary Fig. 56, there were no significant changes in all treatments, showing a favorable immunological safety profile of T@GA-DX-lipo in vivo. These findings demonstrate that the pre-adsorption antibody strategy offers a straightforward and effective platform for targeted delivery, enabling potent cancer therapy.

## Discussion

Monoclonal antibodies and their fragments, celebrated for their high selectivity and affinity, are among the most promising targeting ligands for nanomedicine[7,40,41]. Currently, over ten nanoparticle systems modified with antibody ligands have entered clinical trials. However, most of these systems rely on chemical conjugation for surface modification. This approach, while effective, complicates production and introduces the risk of protein corona formation, which may obscure the targeting function and reduce efficacy[18].

Our study presents a universal approach inspired by the natural binding affinity of polyphenol compounds, such as gallic acid, for proteins. We utilized a GA-lipo system that can controllably adsorb proteins while preserving their activity. Unlike existing physical adsorption techniques, GA-lipo is stable, resisting protein corona effects, simple for large-scale production, and adaptable for various ligand modifications.

To contextualize our strategy with existing HER2-targeted therapies, we compared T@GA-DX-lipo with Enhertu (trastuzumab deruxtecan). Both platforms utilize trastuzumab for targeting and a deruxtecan-derived payload, but differ significantly in design and mechanism. Enhertu relies on covalent conjugation, requiring precise drug-to-antibody ratios, complex manufacturing, and potential structural perturbation of the antibody. In contrast, T@GA-DX-lipo enables modular, non-covalent antibody assembly via galloyl–Fc interactions, which preserves antibody conformation and simplifies preparation. Moreover, ADCs are typically limited to 2–8 payload molecules per antibody, while liposomes can encapsulate thousands of drug molecules. In our system, each trastuzumab molecule delivers ~580 DXdd molecules, greatly enhancing the drug-to-antibody molar ratio and therapeutic potency. Mechanistically, galloyl groups mediate directional antibody adsorption, preserving antigen-binding site orientation even in the presence of protein corona, thereby enhancing tumor-specific delivery and synergy.

Despite these promising findings, several limitations should be considered. First, although the galloyl-mediated adsorption strategy offers a convenient route for antibody functionalization, its non-covalent nature may lead to antibody desorption under dynamic in vivo conditions. While our in vivo data support effective targeting within the experimental window, long-term stability of antibody binding remains to be fully validated. Second, compared to small-molecule antibody–drug conjugates (ADCs) such as Enhertu, the nanoparticle-based system may exhibit distinct intratumoral distribution patterns due to differences in size, diffusivity, and tumor penetration behavior. These differences may influence drug release kinetics and therapeutic outcomes and merit further comparative evaluation. Finally, the translational potential of this platform requires additional validation in large-animal models, including assessments of long-term safety, immunogenicity, and pharmacokinetics under clinically relevant conditions.

Together, this work introduces a generalizable strategy for constructing targeted nanocarriers via non-covalent protein assembly, bridging the precision of ADCs and the flexibility of nanomedicine to support translational advancement in targeted chemotherapy. Our galloylated liposome platform demonstrates potent targeting capabilities, ease of formulation, and high delivery efficiency. This versatile platform holds great promise for overcoming the bottlenecks in the clinical translation of targeted nanomedicines.

## Methods

### Ethical statement

Our research adheres to all relevant ethical guidelines. All animal studies were conducted in compliance with ethical standards approved by the Institutional Animal Care and Use Committee of Shenyang Pharmaceutical University #SYPU-IACUC-S2024-0703-105. Mice were sheltered and maintained in the Center for Comparative Medicine at Northwestern University at 21 °C with 40–60% humidity and a 12 h/12 h dark/light cycle.

### Chemicals and materials

Trastuzumab, cetuximab were purchased from Shanghai Aladdin Biochemical Technology Co., Ltd. Transferrin was purchased from Shanghai Yuanye Bio-Technology Co., Ltd. Rituximab and Nimotuzumab were DXd was purchased from Beijing Solarbio Science & Technology Co., Ltd. from TargetMol Chemicals Inc. Gallic acid monohydrate and other chemical reagents were purchased from Dalian Meilun Biotechnology Co., Ltd. (Dalian, China). Hydrogenated Soybean Phospholipids (HSPC), Cholesterol were purchased from AVT (Shanghai) Pharmaceutical Tech Co., Ltd. Sepharose CL-4B gel, 0.2% phosphotungstic acid, and Hoechst 33,342 were bought from Beijing Solarbio Corporation. Cell culture reagents were purchased from Dalian Meilun Biotechnology Co., Ltd. Fetal bovine serum (FBS), cell culture dishes/plates, and round coverslips were supplied by NEST Biotechnology (Jiangsu, China). Goat Anti-Human IgG Fc (DyLight® 488) (ab97003) was acquired from Abcam. 12 nm Colloidal Gold AffiniPure™ Goat Anti-Human IgG (H + L) (EM Grade) was obtained from Rupesh Nanjunda, Janssen Research & Development, HER2 rabbit IgG (bs-2156R), and F(ab')₂ Fragment Goat Anti-Mouse IgG H&L, PE conjugated (bs-60296G-PE) was purchased from BIOSS. IFN-γ (JL12152), TNF-α (JL10208), and IL-6 (JL14113) ELISA kits were purchased from Shanghai Jianglai Biotech Co., Ltd.

### GA-lipid synthesis

Tert-butyldimethylsilyl (TBS)-protected gallic acid (GA-TBS) was synthesized following the method described in our previous report[42]. GA-modified lipids were synthesized in two steps: first, GA-TBS was esterified with lipids or cholesterol (Chol), followed by hydrogenolysis using a 3.6-fold excess of TBAF to remove the protective groups, exposing the functional phenolic moieties.

Synthesis of GA-lipid: GA-glycerol analogs were synthesized by coupling GA-TBS to a triglyceride skeleton or Chol via ester bonds. Triglyceride skeletons of varying chain lengths and saturation levels were synthesized as per the literature[43]. GA-TBS (512.00 mg, 1 mmol), EDCI (573.10 mg, 3 mmol), and DMAP (366.50 mg, 3 mmol) were dissolved in 100 mL dichloromethane and stirred in an ice bath for 2 h. Then, triglyceride skeletons (2 mmol) were added. The reaction continued at 25 °C for 12 h. The reaction was monitored by thin-layer chromatography (TLC), and the crude derivatives were purified by silica gel chromatography with a 20:1 n-petroleum ether/ethyl acetate solvent system, yielding white solids or oils (70%).

The resulting product (0.7 mmol) was weighed into a 250 mL flask containing 100 mL anhydrous tetrahydrofuran and stirred under an ice bath. TBAF (2.52 mmol) was added dropwise, followed by continuous stirring for 30 min. The mixture was diluted with acetate, washed three times with brine, and the crude product was purified by preparative liquid chromatography (yield: 80%).

Synthesis of GA-PEG-Chol: GA-TBS (512.00 mg, 1 mmol), EDCI (573.10 mg, 3 mmol), and DMAP (366.50 mg, 3 mmol) were dissolved

in 100 mL dichloromethane and stirred under an ice bath for 2 h. Ethylene glycol with varying degrees of polymerization (2 mmol) was then added dropwise. The reaction continued at 25 °C for 12 h and was monitored by TLC. The crude derivatives were purified by silica gel chromatography using a 50:1 n-dichloromethane/methanol solvent system (oil, yield: 65%).

Synthesis of GA-P-COOH: GA-P-OH (0.6 mmol), succinic anhydride (1.2 mmol), and DMAP (144.6 mg, 0.6 mmol) were dissolved in 20 mL dichloromethane and stirred for 24 h. TLC was used to monitor the reaction, and the crude derivatives were purified by silica gel chromatography using a 40:1 n-dichloromethane/methanol solvent system (oil, yield: 60%).

Synthesis of GA-P-Chol: GA-P-Chol was synthesized similarly to GA-lipids, except GA-P-COOH was used in place of GA-TBS.

## DXdd synthesis

DXd (49.35 mg, 0.1 mmol), 4-(4-Methyl-1-piperazinyl) butanoic acid (PA, 89.04, 0.3 mmol), EDCI (172.50 mg, 0.9 mmol), and DMAP (109.94 g, 0.9 mmol) were added to a 50 mL flask. After adding 20 mL DMF, the mixture was reacted at 25 °C for 12 h. The crude product was purified using preparative liquid chromatography (yellow solid, yield: 65%).

The final derivative was characterized using $^1$H NMR (400 MHz, Bruker AV-400), high-resolution mass spectrometry (Agilent 1100 Series LC/MSD Trap), and HPLC analysis.

## Surface-galloylated liposomes preparation and characterization

GA-lipo was prepared via the ethanol inpouring method. Typically, 150 mg HSPC, 40 mg Chol, and 10 mg GA-lipid (40 mg mL$^{-1}$ in ethanol) were injected into 10 mL of 250 mM $(NH_4)_2SO_4$ or 0.6 mol/L SOS-TEA solution. After stirring at 65 °C for 30 min, the ethanol was removed using a rotary evaporator. Vesicle size was reduced using a high-pressure extrusion device. Unencapsulated inner phase solution was removed using a Sepharose CL-4B gel column equilibrated with HBS buffer. Blank liposomes were stored at 4 °C until use. Variants of blank liposomes were prepared by altering the lipid composition or the inner phase solution.

DOX (20 mg/mL in water) or DXdd (20 mg/mL in DMSO) was added dropwise to blank liposomes at 60 °C for 30 min, with a final drug-to-lipid ratio of 1:10 (w/w). Unloaded drugs and DMSO were removed by gel filtration using a Sepharose CL-4B column. Drug-loaded liposomes were stored in the dark at 4 °C.

DiR/DiO-labeled liposomes were prepared via the thin-film hydration method. In brief, 150 mg HSPC, 38 mg Chol, 10 mg GA-lipid, and 2 mg DiR/DiO were dissolved in chloroform. A uniform thin film was formed by rotary evaporation at 37 °C. The film was hydrated with 10 mL of 250 mM $(NH_4)_2SO_4$ at 65 °C for 30 min. Vesicle size reduction and removal of unencapsulated solution were performed as described previously and stored at 4 °C.

Antibody-liposome conjugates were prepared via maleimide-thiol coupling. Typically, 150 mg HSPC, 40 mg Chol, and 10 mg Chol-Maleimide, 2 mg DiO or DiR (40 mg/mL in ethanol) were dissolved in chloroform. A uniform thin film was formed by rotary evaporation at 37 °C. The film was hydrated with 10 mL of 250 mM $(NH_4)_2SO_4$ at 65 °C for 30 min. Vesicle size reduction and removal of unencapsulated solution were performed as described previously and stored at 4 °C. Mal-lipo were incubated with thiol-modified trastuzumab (molar ratio antibody:maleimide = 1:3) at 4 °C for 12 h. Unconjugated antibody was removed by ultrafiltration (MWCO 300 kDa), and the final liposomes were characterized for particle size, PDI, and zeta potential using dynamic light scattering.

## X-ray photoelectron spectroscopy (XPS)

XPS data acquisition was conducted using a Thermo Escalab 250Xi system equipped with a monochromated AI Kα (hv = 1486.6 eV), operated at 150 W (4.8 kV, 1.6 A).

## Preparation of protein@GA-lipo via protein adsorption

GA-lipo (10 mg/mL) was added to the protein solution (10 mg/mL in HBS or buffers of different pH) and incubated at 25 °C for 1 h. After centrifugation, the liposome pellet was collected and resuspended in HBS, with washing repeated three times. The final protein@GA-lipo formulation was suspended in HBS.

## Preparation of Trf@GA-lipo via protein adsorption

GA-lipo (10 mg/mL) was added to a Trf solution (10 mg/mL in HBS) and incubated at 25 °C for 1 h. After centrifugation and washing with HBS buffer three times, the final Trf@GA-lipo formulation was suspended in HBS. The Trf concentration for in vitro experiments was regulated by incorporating varying molar percentages of total lipids into each sample (0.005%, 0.025%, 0.050%, 0 mol%), corresponding to final well concentrations of 2.5, 12.5, and 25.0 ng/mL, respectively.

## Preparation of T@GA-lipo via protein adsorption

GA-lipo (10 mg/mL) was added to the TRA solution (10 mg/mL in HBS) and incubated at 25 °C for 1 h. After centrifugation and washing with HBS buffer three times, the final T@GA-lipo formulation was suspended in HBS. TRA concentration for in vitro experiments was set at 0.025 mol%, corresponding to a final well concentration of 25.0 ng/mL.

## Size and zeta potential measurement

Particle size, polydispersity index (PDI), and zeta potential of liposomes were measured in triplicate using a Zetasizer (Nano ZS, Malvern, UK). The results were derived from three independent parallel samples.

## Drug loading and release profiling

The quantities of DOX and DXdd loaded in liposomes were measured by high-performance liquid chromatography (Chromaster & 5610). For the release study, liposomes were placed in a 10,000 KDa dialysis membrane and placed in PBS at 37 °C. The results were derived from three independent parallel samples.

## Protein adsorption efficiency

Adsorption efficiency was determined by ultracentrifugation. Proteins were incubated with FITC in carbonate buffer solution at 4 °C for 8 h. The FITC-labeled protein was first purified by dialysis at 4 °C for 12 h and followed by a purification with Sephadex-based size exclusion chromatography (SEC) to completely remove the residual free FITC. FITC-labeled proteins were incubated with different liposomes to prepare protein@GA-lipo. Unbound proteins were first removed by centrifugation at 20,000 × g for 30 min. The concentrations of unbound and total protein were measured by a microplate reader, and adsorption efficiency was calculated as: Adsorption efficiency (% = 1 − C$_{uncoated}$ protein/C$_{total protein}$×100%).

## Transmission electron microscopy

The morphology of liposomes was observed via transmission electron microscopy (TEM, 100CX II, Japan). 10 μL of liposomes were dropped onto a 200-mesh carbon-coated copper grid and allowed to sit for 5 min, followed by staining with 0.2% phosphotungstic acid for 30 s.

## Tryptophan quenching assay

GA-lipo or lipo (10 mg/mL) was incubated with proteins (2.5 mg/mL Trf, 5 mg/mL TRA) at 25 °C for 2 h. A microplate reader was used to record tryptophan fluorescence ($\lambda_{Ex}$ = 280 nm, $\lambda_{Em}$ = 300–450 nm). To assess the interaction strength between proteins and GA-lipo, the fluorescence of Trf@GA-lipo and buffer with liposomes (background) was measured after incubation with 100 mM Triton X-100, SDS, NaCl, or urea for 4 h. Tryptophan quenching recovery was determined by normalizing the area under the curve (AUC) to folded Trf, after

subtracting the background fluorescence. Experiments were performed in triplicate.

## Immunogold labeling

Liposome samples (10 mg/mL) were mixed with TRA (0.025 mol%) and incubated at 25 °C for 1 h. After centrifugation, the unbound antibody was removed, and the liposomes were resuspended in PBS, washed three times. 10 μL of the sample was diluted with 90 μL PBS and incubated with 20 μL of secondary gold-coupled antibodies (1% BSA in PBS). After centrifugation and three washes to remove uncoupled secondary antibodies. Visualization was performed using TEM (100CX II, Japan).

## Molecular dynamics simulations

The initial antibody was constructed by Discovery Studio (version 2023). All disulfide bonds were correctly paired in the constructed antibody. And then, this initial structure was used as the initial structure for the dynamic simulation. The same as GA-P and HEPES were optimized using the DFT method by B3LYP/def2-TZVP level with the ORCA program. Molecular dynamics (MD) simulations were conducted using GROMACS (version 2023.4). Twenty GA-P (GA-P1-Chol) molecules were randomly inserted into the $13.475 \times 12.932 \times 16.523$ nm box, followed by solvation with TIP3P water. Adding counterions keeps the system electrically neutral. The system employed was a mixture of 20 mM HEPES and 70 mM NaCl, which provided a suitable buffering environment for the subsequent simulation studies, helping to maintain the structural and functional characteristics of biomolecules and thereby ensuring the reliability and scientific validity of the simulation results. The initial system was subjected to energy minimization via the steepest descent algorithm for 50,000 steps. Subsequently, the system was used in a restrained pre-equilibrated under the NVT ensemble for 500 ps to raise the temperature to 298.15 K. Afterward, a 500 ps NPT restrained pre-equilibration was performed to adjust the pressure to 1 atm. Once the temperature and the pressure were stable, the restraints were removed to allow the antibody to be within the solvent environment. The system was maintained under constant pressure using the C-rescale coupling method, with a cutoff radius of 1.0 nm and a time step of 1 fs. The total simulation time was 100 ns. The antibody was used Amber99SB-ILDN force field, and the GA-P and HEPES were generated with the Gaff force field using the sobtop program, with resp charge calculated using the Multiwfn program. The tip3p model was applied for water molecules. The V-rescale thermostar and C-rescale barostat were employed to control the system temperature and pressure, respectively. Short-range electrostatic and van der Waals interactions were calculated with a 1.0 nm cutoff, while long-range electrostatic interactions were computed using the particle mesh Ewald (PME) method. The LINCS algorithm constrained bonds involving hydrogen atoms, and periodic boundary conditions (PBC) were applied in all three dimensions of the simulation box.

## Isothermal titration calorimetry (ITC)

ITC was performed to evaluate the interaction between trastuzumab and GA-lipo. After degassing all samples for 10 min, 300 μL of trastuzumab (2 μM) was loaded into the sample cell, and 60 μL of GA-lipo (20 μM, lipid equivalent) was placed in the syringe. Titration was carried out at 25 °C with 20 injections at 200 s intervals. Thermodynamic parameters (KD, n, ΔH, ΔS) were analyzed using NanoAnalyzer software.

## Detection of antibodies on liposome surfaces by flow cytometry

Liposomes (1 mg) were incubated with TRA/Cet (0.05 mg) to allow protein adsorption. TRA was labeled with FITC prior to incubation to detect its presence on the liposome surface. After washing and resuspending in PBS, the liposomes were detected by flow cytometry. FITC-positive liposomes were quantified by defining blank liposomes

as false positives. The gating strategy for flow cytometry (BD FACS-Calibur) is presented in Supplementary Fig. 57.

A PE-labeled secondary antibody specific to the mouse Fab fragment (1:50 dilution, BIOSS) was employed to identify the functional Fab region on the surface of the liposomes. After washing and resuspending in PBS, the liposomes were detected by flow cytometry (BD FACSCalibur). PE-positive liposomes were quantified by defining blank liposomes as false positives. The gating strategy for flow cytometry is presented in Supplementary Fig. 58.

To determine antibody orientation on the liposome surface, secondary Alexa Fluor-488-labeled anti-mouse Fc fragment-specific antibody (1:50 dilution, Abcam) was used. After resuspension in PBS, liposomes were analyzed by flow cytometry (BD FACSCalibur), and Alexa Fluor-488-positive liposomes were quantified by defining blank liposomes as false positives. The gating strategy for flow cytometry is presented in Supplementary Fig. 59.

## Detection of antibodies on liposome surfaces in FBS stability studies

TRA was labeled with FITC and incubated with liposomes at 25 °C for 1 h. The liposomes were then incubated in 50% FBS at 37 °C At various time points, unbound protein was removed by centrifugation at $20,000 \times g$ for 30 min. The concentrations of unbound and total protein were measured using a microplate reader. The results originated from independent parallel samples.

Secondary PE-labeled anti-mouse Fab fragment-specific antibody (BIOSS) was used to detect the Fab region of the antibody on the liposome surface. After washing and resuspending in PBS, the liposomes were detected by flow cytometry (BD FACSCalibur). PE-positive liposomes were quantified by defining lipo-functionalized nanoparticles as false positives.

## Cells culture and assays

Cells were obtained from Wuhan Procell Life Technology Co., Ltd. 4T1 cells (CL-0007) and THP-1 cells (CL0213) were cultured in RPMI-1640 supplemented with 10% FBS. SKOV3 cells (CL0215) were grown in McCoy's 5A medium supplemented with 10% FBS, and MCF-7 cells (CL-0619) were maintained in Dulbecco's Modified Eagle Medium (DMEM) with 10% FBS.

## Cellular uptake studies of drug-loaded liposomes

Around 20,000 4T1 cells were plated in 12-well plates and incubated for 24 h. The cells were incubated with different liposomes (5 μg/mL DOX) for various durations (4, 8, 12, and 24 h) in culture medium containing 10% FBS. At the designated time points, cells were collected, resuspended in 400 μL PBS. The DOX uptake was quantified by precipitating the protein and measuring the fluorescence using a microplate reader. Protein content was quantified using BCA kits. Experiments were conducted in triplicate.

Similarly, 50,000 SKOV3 cells were seeded in 12-well plates. After culturing for 24 h, cells were incubated with different liposomes (5 μg/mL DXdd) for various durations (4, 8, 12, and 24 h) in culture medium containing 10% FBS. To examine the effect of varying FBS concentrations, cells were cultured in McCoy's 5A medium with different HS concentrations (0%, 10%, and 50%). At designated time points, cells were collected, resuspended in 400 μL PBS, and DXdd uptake was quantified using the same method as for DOX. Protein content was measured using BCA kits. Experiments were conducted in triplicate.

To compare the targeted capacity of adsorption and chemical conjunction strategies. Cells were then incubated with DiO-labeled liposomes (GA-lipo, Mal-lipo, T@GA-lipo, and T@Mal-lipo, equivalent to 5 μg/mL DiO) for 12 h. To assess serum interference, parallel experiments were performed in media containing 0%, 10%, or 50% FBS. At designated time points, cells were washed with cold PBS, collected,

and resuspended in 400 μL PBS. Fluorescence intensity was measured using a flow cytometer (BD FACSCalibur).

## Cellular uptake in THP-1-derived macrophages

DiO-labeled liposomes (0.5 mg/mL) were mixed with 0.025 mol% TRA (or isotype control, 1 mg/mL) in HBS for 2 h at 25 °C and washed three times. Human monocytic THP-1 cells were seeded in 12-well plates at a density of $1 \times 10^6$ cells/well and treated with 100 ng/mL phorbol 12-myristate 13-acetate (PMA, Sigma) for 48 h to induce macrophage differentiation. Following differentiation, cells were incubated with DiO-labeled liposomes (5 μg/mL) in RPMI-1640 supplemented with 10% FBS for 12 h at 37 °C. After incubation, cells were washed three times with ice-cold PBS, harvested by mild trypsinization (0.25% trypsin–EDTA, 2 min), and resuspended in PBS for flow cytometric analysis using a BD Accuri C6 Plus cytometer. The percentage of DiO-positive cells was quantified to evaluate non-specific liposome uptake. All experiments were performed in triplicate. After 12 h incubation with different liposomal formulations under the same conditions described above, the cell culture supernatants were collected, measured the concentrations of TNF-α, IL-6, and IL-10 were measured using commercial ELISA kits. All assays were conducted in biological triplicate.

## Colocalization experiment

DiR-labeled liposomes (0.5 mg/mL) were mixed with 0.025 mol% TRA (or isotype control, 1 mg/mL) in HBS for 2 h at 25 °C and washed three times. SKOV3 cells were incubated with the liposomes (5 μg/mL) at 37 °C for 8 h. The cells were then stained with LysoTracker Red for 30 min at 37 °C and Hoechst 33258 for 10 min. Colocalization images were obtained using a TCS SP2/AOBS CLSM confocal microscope (Nikon, ×60 objective magnification). MCF-7 cells (HER2-negative) were used as a control and cultured in DMEM.

## Antibody blocking experiment

To validate receptor-mediated uptake, SKOV3 tumor cells were seeded in 12-well plates at $5 \times 10^4$ cells/well and allowed to adhere for 24 h. Cells were pre-incubated with an excess of anti-HER2 antibody (10 μg/mL, trastuzumab) in McCoy's 5 A medium containing 10% FBS for 30 min at 37 °C to block HER2 receptors. Subsequently, DiO-labeled T@GA-lipo (5 μg/mL) was added and incubated for an additional 8 h under the same conditions. After treatment, cells were washed thoroughly with PBS, harvested by trypsinization, and analyzed by flow cytometry (BD FACSCalibur) to determine the extent of liposome uptake. All experiments were performed in triplicate.

## SDS-PAGE Analysis for Coated Protein in the Hard Protein Corona

Liposomes (5 mg) were incubated with 50% rat serum at 37 °C for different times (5 min, 15 min, 30 min, 1 h, 2 h, 6 h, and 24 h). Hard corona purification was performed by centrifugation (18,000 × *g*, 30 min) followed by three washes with PBS. The pellet was resuspended in 0.5 mL PBS and quantified using BCA kits.

50 μL sample and 5 μL SDS-PAGE loading buffer were mixed and heated at 95 °C for 5 min. After centrifugation at 18,000 × *g* for 5 min to remove liposomes, the supernatant containing desorbed hard corona proteins was analyzed. Diluted rat serum (1:1000) and Trf or TRA (0.5 μg) were used as controls. Protein separation was performed on vertical PAGE precast gels, followed by staining with 0.25% Coomassie Brilliant Blue for 30 min. After rinsing for 30 min, gels were imaged using the ChemiDocTM imaging system (Supplementary Figs. 60–62).

## Cell viability assay

Liposomes were incubated with 2000 4T1 cells at varying concentrations and incubated at 37 °C. Five replicates were performed for each measurement. After 48 h, MTT (20 μL) was added to the cells and

incubated for 4 h, followed by dissolving with DMSO, and absorbance was measured at 570 nm.

## Pharmacokinetics assay

Male SD rats (-180–220 g, 6–8 weeks) were intravenously administered different liposome formulations (2.5 mg/kg DOX equivalent, *i.v.*). At determined times, blood samples were collected and centrifuged to obtain plasma. Plasma concentrations of DOX were measured using a microplate reader.

## In vivo biodistribution studies

Female BALB/c mice (18–22 g, 6–8 weeks) bearing 4T1 tumors were used to investigate the accumulation of DOX-loaded liposomes. The accumulation of DOX-loaded liposomes (5 mg/kg DOX, *i.v.*) in tumors and major organs was analyzed 2, 6, 12, 24, 48 h post-administration using a microplate reader.

Female BALB/c nude mice (18–22 g, 4–6 weeks) bearing subcutaneous SKOV3 tumors (-200 mm³) were used to investigate the accumulation of DXdd-loaded liposomes. The accumulation of DiR-labeled T@GA-lipo (1 mg/kg DiR) in tumors (0.1–0.3 g) was analyzed using in vivo imaging systems on the SKOV3 xenograft model. After 48 h post-administration, major tissues were collected to assess biodistribution. The accumulation of DXdd-loaded liposomes (5 mg/kg DXdd, *i.v.*) in tumors and major organs was analyzed 2, 6, 12, 24, 48 h post-administration using HPLC.

Female BALB/c nude mice (18–22 g, 4–6 weeks) bearing subcutaneous SKOV3 tumors (-200 mm³) were intravenously injected with DiR-labeled liposomes (GA-lipo, Mal-lipo, T@GA-lipo, and T@Mal-lipo, DiR dose: 1 mg/kg, *i.v.*). At 1, 4, 8, 12, 24, 36, and 48 h post-injection, mice were imaged using an IVIS Spectrum imaging system (IVIS Spectrum) to monitor whole-body distribution. At 48 h, mice were euthanized, and major organs (heart, liver, spleen, lung, kidney, and tumor) were collected for ex vivo imaging. Fluorescence intensity in each organ was quantified using Living Image software.

## Efficacy study

To assess the antitumor effect of Trf@GA-DOX-loaded liposomes, 4T1 cells ($1 \times 10^6$ cells in 100 μL PBS) were injected into the right flank of female BALB/c mice (-18–22 g, 6 weeks). When tumors reached -100 mm³, mice were treated with drugs and intravenously administered every 3 days (3 mg/kg DOX equivalent, *i.v.*) for a total of 3 doses. Animals were randomly assigned to groups.

Similarly, SKOV3 cells ($5 \times 10^6$ cells in 100 μL PBS containing 5% Matrigel) were inoculated into the underarm of female BALB/c-nu mice (-18–22 g, 6 weeks). When tumors reached -100 mm³, mice were treated with DXd formulations and administered intravenously every 7 days (5 mg/kg DXdd equivalent, *i.v.*) for a total of 3 doses. After the treatment period, mice were sacrificed, and tumors and major organs were collected for H&E analysis. Tumors were also processed for immunofluorescence staining (TUNEL, Ki67, and HER2).

## In vivo dose selection study

To determine the appropriate dose for therapeutic evaluation, female BALB/c-nu mice (6–8 weeks old) were administered free DXd solution or DX-lipo at increasing doses (3, 5, or 10 mg/kg) via tail vein injection every three days for a total of three doses. Mice were monitored for signs of weight change.

## Blood chemistry analysis

After intravenous administration of different DXdd formulations for four doses, -300 μL of peripheral blood was collected and analyzed using the Mindray BC-5000 Vet. Serum was obtained by centrifugation of blood samples, and hepatorenal function and Cytokine analysis were evaluated.

## Statistics and reproducibility

Quantitative results are presented as mean ± standard deviation (s.d.). Statistical significance for two experimental samples was determined using Student's unpaired t-test. For comparisons among multiple treatment groups, one-way analysis of variance (ANOVA) with Tukey's post-hoc test was applied. Data are significantly different if $P < 0.05$. The specific statistical methods were indicated in the figure legend. GraphPad Prism was used for statistical analysis.

## Reporting summary

Further information on research design is available in the Nature Portfolio Reporting Summary linked to this article.

## Data availability

All the data generated in this study are provided in the article, Supplementary Information, Source Data file, and deposited in the figshare database under accession code [https://doi.org/10.6084/m9.figshare.29382998]. Source data is available for Figs. 2–6 and Supplementary Figs. 1–49 in the associated source data file. Source data are provided with this paper.

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

## Acknowledgements

This work was financially supported by the National Natural Science Foundation of China (Nos. 82273878 and 82473872 to Y.W.) and Shenyang Municipal Natural Science Foundation (23-503-6-11 to H.L.).

## Author contributions

J. Li, Y. Wang, J. Yu, H. Liu, and Z. He conceived and designed the project. J. Li, J. Yu, J. Song, Y. Zhang, N. Li, Z. Wang, M. Qin, M. Zhao, B. Zhang, R. Huang, S. Zhou, and D. Liu performed experiments and data analysis. All authors participated in drafting the manuscript, engaging in discussions on the results and their implications, and revising the paper throughout its development.

## Competing interests

The authors declare no competing interests.
