## [Transparent Peer Review file · Nature Communications]

Galloylated Liposomes Enable Targeted Drug Delivery by Overcoming Protein Corona Shielding

Corresponding Author: Professor Yongjun Wang

Version 0:

Reviewer comments:

Reviewer #1

(Remarks to the Author)

This manuscript presents a novel approach to overcoming the protein corona effect in targeted drug delivery by developing galloylated liposomes (GA-lipo), which enable stable and controlled adsorption of monoclonal antibodies through non-covalent interactions. The study leverages galloyl-modified lipids, which have a strong affinity for the Fc region of antibodies, allowing for optimal ligand orientation while preventing shielding by the protein corona. By incorporating trastuzumab as a model antibody and DXd derivatives as a cytotoxic payload, the authors demonstrate that GA-lipo can efficiently adsorb antibodies without chemical conjugation, achieve high drug loading efficiencies (~95%), and maintain targeting functionality even after exposure to biological environments. In vitro and in vivo studies confirm that GA-lipo significantly enhances tumor targeting and therapeutic efficacy, reducing tumor growth by up to 8-fold compared to non-targeted controls, while maintaining stability and safety.

The study systematically characterizes GA-lipo in terms of liposome composition, protein adsorption efficiency, pharmacokinetics, and therapeutic outcomes. Using XPS, fluorescence quenching, and flow cytometry, the authors validate the antibody orientation and demonstrate that the Fab region remains exposed despite protein corona formation, ensuring effective receptor binding. Comparative tumor models confirm that GA-lipo achieves higher drug accumulation in tumors, reduced off-target effects, and improved treatment outcomes compared to traditional liposomal formulations. The manuscript highlights broad applicability, showing that GA-lipo can be extended to other monoclonal antibodies (e.g., cetuximab) and various therapeutic payloads. The authors propose that GA-lipo represents a scalable and clinically translatable solution for antibody-functionalized nanomedicines, offering a more efficient and reproducible alternative to conventional chemical conjugation techniques.

The study is well-conducted and has robust experimental validation across multiple models, including in vitro and in vivo assessments. The authors convincingly demonstrate the advantages of GA-lipo in preserving ligand orientation, preventing protein corona masking, and improving tumor targeting. Given its novelty, significance, and broad applicability in nanomedicine, the study is highly suitable for Nature Communications. However, major revisions are necessary to enhance mechanistic insights, comparative performance validation, and stability assessments.

1) We like to focus on the mechanistic clarity on galloyl-antibody interactions. The authors propose that hydrophobic interactions between galloyl groups and Fc regions drive antibody adsorption. However, electrostatic interactions, hydrogen bonding, and π - π stacking could also contribute. The study relies on XPS and fluorescence quenching but lacks quantitative or molecular-level evidence to confirm the interaction type. The Fc region is known to interact with multiple biomolecules; thus, the specificity of galloyl for Fc needs stronger justification.

2) Given the significance of the study, we share some concerns about the platform's generalizability if it is universally applicable to most or all monoclonal antibodies. If GA-lipo is limited to antibodies with certain properties (e.g., specific isoelectric points, hydrophobic domains), its broad applicability could be questioned. We would like to see additional examination on 2 to 3 more antibodies to validate the platform's versatility and provide data on the formulation reproducibility in terms of batch consistency in adsorption efficiency and particle stability.

3) While the authors claim GA-lipo outperforms conventional chemical conjugation, no direct side-by-side comparison is included. Covalent conjugation methods (e.g., maleimide-thiol, click chemistry) are widely used in clinically approved

nanomedicines. This study would be more compelling if it provided quantitative performance metrics (e.g., targeting efficiency, stability, in vivo tumour accumulation) compared to covalent antibody-liposome conjugates, as it could allow the industry to adopt such unique approaches.

4) The manuscript states that GA-lipo is stable at 4°C for six weeks, but this does not address long-term storage stability, which is crucial for future clinical translation. Stability in biological fluids (e.g., plasma, whole blood) beyond 24–48 hours has not been tested. An evaluation of the liposome integrity in human plasma over 7+ days to simulate clinical pharmacokinetics. We would also like to see long-term storage stability data at different temperatures (e.g., -20°C, 25°C) over at least three months. Additional DLS and Zeta potential measurements over time would provide insight into nanoparticle aggregation and degradation.

5) In the supplementary section, the pharmacokinetics (Supplementary Table 4, 5) provides valuable insight into liposome circulation time and tumor accumulation, but lacks long-term clearance data. The biodistribution study (Supplementary Fig. 22, 35) is well-executed, but it would be useful to track organ-specific clearance over time.

6) The cell viability assays (Supplementary Fig. 25a, 25b) demonstrate GA-lipo's targeted delivery, but non-specific uptake by Fc receptor-expressing immune cells is not assessed. (e.g., cytokine response) is not explored. We would like to see control experiments that validate that GA-lipo is receptor-mediated endocytosed instead of passive endocytosis.

7) The in vivo tumor model (Supplementary Fig. 36, 37) confirms therapeutic efficacy, yet systemic immune activation (e.g., cytokine response) is not explored. Perform cytokine analysis (IL-6, TNF- α , IFN- γ) in serum samples from treated mice to check for systemic immune activation or suppression.

8) In Figure 5f, the dosage of GA-lipo administered in the mouse treatment experiment was 5 mg/kg. Please explain the rationale behind selecting this dosage and whether it reflects preclinical optimization or other considerations. Additionally, there is an inconsistency regarding the tumor inoculation method. The figure caption states that SKOV3 cells were inoculated orthotopically, but Figure 5f and the methods section indicate subcutaneous inoculation into the underarm of female BALB/c-nu mice. Please clarify the correct method of tumor inoculation.

9) It is known that galloyl acid has tumor inhibitory properties. In Figure 5A, the cytotoxicity of GA-DXdd-lipo appears to be higher than that of DXdd-lipo, despite similar release behaviors between the two formulations. The authors should discuss whether the galloyl acid modification contributes directly to the enhanced therapeutic effect observed.

10) Inspired by Enhertu, a clinically approved ADC, the authors designed T@GA-DX-lipo. However, the advantages of T@GA-DX-lipo over Enhertu are not fully elaborated in the manuscript. It is recommended to include a more detailed comparison between the two platforms, highlighting the unique benefits of T@GA-DX-lipo. Additionally, the discussion section would benefit from an objective analysis of the study's limitations and the potential challenges associated with translating these findings into clinical applications.

Reviewer #2

(Remarks to the Author)

Li et al. presents an interesting surface modification of liposomes via incorporation of gallic acid-modified lipids, allowing stable noncovalent binding of antibodies to the surfaces of the nanoparticles. The manuscript is overall clearly written and the therapeutic results are convincing in a cancer model of Dox delivery. Publication is recommended after some modifications/additions. Mainly, additional details are needed for methods and figures, as well as additional methods of characterizing protein adsorption to the liposome surfaces. Suggested changes are outlined below:

Major issues:

- 1) Overall, experimental details need to be described in more detail in the results and in the figure captions. In many cases, the route of administration nor concentrations of administered agents is described. These should be clearly stated in both locations so that the reader can better interpret the data.
- 2) The experiments characterizing the orientation of antibodies are not convincing and thus do not support their conclusion that the Fc region of IgG1 primarily complexes with galloyl moieties on GA-P1-lipo. Their experiments assume only two possible orientations of the antibody, while numerous are possible that could account for difficulty in the anti-FC antibodies binding. There could be diverse denatured structures in between. Alternative methods of characterization/validation are necessary or their conclusions should be updated in the text to be less confident.
- 3) Overall, there is insufficient characterization of what is binding to the nanoparticle surfaces, their structure, and their stability. It was odd that a 70% binding efficiency was achieved regardless of the protein concentration in the solution during loading. Are all accessible spaces bound? Many noncovalent techniques for protein binding use a "filler/blocking" protein to cover locations that are not bound by the desired antibodies. Is this necessary here? Do proteins from serum subsequently bind to the nanoparticle surfaces if these additional locations are not blocked? Does the amount of incorporated gallic acid modified lipids correlate with the amount of bound protein?
- 4) Circular dichroism analysis could help assess changes in secondary structure upon binding of proteins and antibodies. It is not clear if denaturation occurs at the nanoparticle surface.
- 5) After antibody binding, what proteins can subsequently bind to the nanoparticle surfaces upon exposure to serum? Do proteins exchange over time (i.e. Vroman effect)?

Minor issues:

- 6) Several grammatical errors were found throughout the text. Thorough reread and editing is recommended

7) Many of the figures, especially in the supplement are a) not referenced in the text or b) referenced out of order. For example, Supplementary figs 9-12 are not discussed and 13a and 13b are out of order.

Version 1:

Reviewer comments:

Reviewer #1

(Remarks to the Author)

the authors have addressed my concerns and the manuscript is now acceptable for publication

Reviewer #2

(Remarks to the Author)

Authors have thoroughly addressed all issues that were raised and provided interesting and supportive new data that improves the manuscript. Publication of the manuscript in its current form is recommended.

Dear reviewers,

We are truly grateful to your valuable comments and hard work when handling with our manuscript entitled “Galloylated Liposomes: Overcoming Protein Corona Shielding for Enhanced Targeted Drug Delivery” (NCOMMS-25-02797-T). Based on the comments and suggestions, we have made further modifications on the original manuscript. The corrected portions in the revised manuscript have been highlighted in red. Below we summarize a point-by-point response to each of the comments from the reviewers.

We appreciate the critical reviews of the manuscript and hope the revised version attached is improved to a significant degree to merit further consideration of publication.

Response to reviewers:

Reviewer #1 (Remarks to the Author):

This manuscript presents a novel approach to overcoming the protein corona effect on targeted drug delivery by developing galloylated liposomes (GA-lipo), which enable stable and controlled adsorption of monoclonal antibodies through non-covalent interactions. The study leverages galloyl-modified lipids, which have a strong affinity for the Fc region of antibodies, allowing for optimal ligand orientation while preventing shielding by the protein corona. By incorporating trastuzumab as a model antibody and DXd derivatives as cytotoxic payload, the authors demonstrate that GA-lipo can efficiently adsorb antibodies without chemical conjugation, achieve high drug loading efficiencies (~95%), and maintain targeting functionality even after exposure to biological environments. In vitro and in vivo studies confirm that GA-lipo significantly enhances tumor targeting and therapeutic efficacy, reducing tumor growth by up to 8-fold compared to non-targeted controls, while maintaining stability and safety.

The study systematically characterizes GA-lipo in terms of liposome composition, protein adsorption efficiency, pharmacokinetics, and therapeutic outcomes. Using XPS, fluorescence quenching, and flow cytometry, the authors validate the antibody orientation and demonstrate that the Fab region remains exposed despite protein corona

formation, ensuring effective receptor binding. Comparative tumor models confirm that GA-lipo achieves higher drug accumulation in tumors, reduced off-target effects, and improved treatment outcomes compared to traditional liposomal formulations. The manuscript highlights broad applicability, showing that GA-lipo can be extended to other monoclonal antibodies (e.g., cetuximab) and various therapeutic payloads. The authors propose that GA-lipo represents a scalable and clinically translatable solution for antibody-functionalized nanomedicines, offering a more efficient and reproducible alternative to conventional chemical conjugation techniques.

The study is well-conducted and has robust experimental validation across multiple models, including *in vitro* and *in vivo* assessments. The authors convincingly demonstrate the advantages of GA-lipo in preserving ligand orientation, preventing protein corona masking, and improving tumor targeting. Given its novelty, significance, and broad applicability in nanomedicine, the study is highly suitable for Nature Communications. However, major revisions are necessary to enhance mechanistic insights, comparative performance validation, and stability assessments.

1) We like to focus on the mechanistic clarity on galloyl-antibody interactions. The authors propose that hydrophobic interactions between galloyl groups and Fc regions drive antibody adsorption. However, electrostatic interactions, hydrogen bonding, and π - π stacking could also contribute. The study relies on XPS and fluorescence quenching but lacks quantitative or molecular-level evidence to confirm the interaction type. The Fc region is known to interact with multiple biomolecules; thus, the specificity of galloyl for Fc needs stronger justification.

Response: We thank the reviewer for pinpointing this important issue. Consistent with the suggestion, we performed 100-ns all-atom molecular-dynamics (MD) simulations to dissect the binding mechanism between galloyl-cholesterol (GA-P1-Chol) and trastuzumab (TRA). The new data have been inserted on page 13 of the main text and in Figure 3g-j and Supplementary Figures 27–28. Key findings are summarized below:

- Multiple non-covalent forces cooperate.

Energy decomposition (Figure 3i) showed that both Coulombic (electrostatic) and Lennard-Jones (van-der-Waals, including hydrophobic) interactions stabilize the complex, while hydrogen-bond counts steadily rise during the trajectory (Figure 3h). π - π stacking is negligible because GA-P1-Chol contains a single phenyl ring, yet its contribution is implicitly captured in the van-der-Waals term.

- Galloyl preferentially associates with the Fc region.

As illustrated in Figure 3g and quantified in Figure 3j, the Fc domains (domains 5–8) display markedly lower total interaction energies than either Fab arm, indicating higher affinity. Per-domain contact maps (Supplementary Fig. 28b-c) reveal dense hydrophobic pockets together with recurrent H-bonds (e.g., D633, E434, T301) unique to Fc, rationalizing the selectivity.

- Antibody conformation remains intact.

RMSD and SASA profiles (Supplementary Fig. 27a-b) show <0.6 nm deviation for individual domains and stable solvent accessibility, confirming that adsorption does not distort the tertiary structure—an essential prerequisite for preserving antigen recognition.

Collectively, these quantitative MD data corroborate our initial hypothesis: hydrophobic contacts constitute the dominant driving force, complemented by hydrogen bonding and electrostatic interactions, yielding a net preference for the Fc region. The revised manuscript explicitly discusses these points and cites the new figures.

Supplementary Fig.27 Conformational stability of trastuzumab during MD simulation. a,

Time-evolution of backbone RMSD (root-mean-square deviation) for the full antibody (black) and its sub-domains (Fab1, Fab2, hinge, Fc). b, Solvent-accessible surface area (SASA) of the full antibody as a function of time.

Supplementary Fig.28 Molecular interaction analysis between trastuzumab and GA-P1-Chol based on molecular dynamics simulations. a, Overall docking model of trastuzumab and GA-P1-Chol, showing the spatial location of eight identified binding domains (domains 1-8) across Fab1, Fab2, and Fc regions. GA-P1-Chol is shown in magenta; individual antibody chains are colored distinctly. b, Representative close-up views of GA-P1-Chol interactions within each domain. Key residues involved in hydrogen bonding (yellow dashed lines) or hydrophobic interactions are labeled. Domain1 (D658, S660): 5 hydrogen bonds; Domain2: hydrophobic interactions; Domain3 (H692, L413): 2 hydrogen bonds; Domain4 (D633, S632, A501): 4 hydrogen bonds; Domain5 (E434): 2 hydrogen bonds; Domain6: hydrophobic interactions; Domain7 (R20): 1 hydrogen bond; Domain8 (T301, D244): 2 hydrogen bonds. c, Binding free energy decomposition of GA-P1-Chol to each domain based on per-residue energy contribution analysis. Bar graphs show the energy

contribution (kcal/mol) of key residues; negative values indicate favorable interactions.

Figure 3 g–j. Molecular-dynamics analysis of GA-P1-Chol binding to trastuzumab. g, Simulation snapshot of an equilibrated TRA protein bound to twenty GA-P1-Chol molecules (in purple). The ligand–protein interfaces are stabilized by hydrophobic pockets and hydrogen-bond networks, as indicated from domain 1 to domain 8, maintaining the structural integrity of TRA. h, Time evolution of hydrogen bonds formed between GA-P1-Chol and the full-length trastuzumab (black) or its Fc region (red) during a 100 ns MD simulation. The increasing trend indicates progressive stabilization of the complex over time, with a higher number of hydrogen bonds observed in the full antibody structure. i, Decomposition of binding energy contributions between GA-P1-Chol and each domain (1–8), showing Coulombic (electrostatic) and Lennard-Jones (van der Waals) interactions. j, Time-resolved total interaction energy (kcal/mol) between GA-P1-Chol and individual antibody domains during the last 20 ns of MD simulation.

2) Given the significance of the study, we share some concerns about the platform's generalizability if it is universally applicable to most or all monoclonal antibodies. If GA-lipo is limited to antibodies with certain properties (e.g., specific isoelectric

points, hydrophobic domains), its broad applicability could be questioned. We would like to see additional examination on 2 to 3 more antibodies to validate the platform's versatility and provide data on the formulation reproducibility in terms of batch consistency in adsorption efficiency and particle stability.

Response: We thank the reviewer for this insightful suggestion. To validate the generalizability of our GA-lipo platform across antibodies with different physicochemical properties, we further examined two additional therapeutic monoclonal antibodies—nimotuzumab (pI \approx 7.5) and rituximab (pI \approx 9.1)—in addition to trastuzumab (pI \approx 8.8) and cetuximab (pI \approx 8.7). Comprehensive characterization was performed, and results are summarized in Supplementary Figures 44–46 and Supplementary Table 7. All results have been integrated into the revised manuscript (page 18).

- High adsorption capacity across different mAbs

After 1 h incubation at 25 °C, GA-P1-lipo exhibited high adsorption efficiency for nimotuzumab (64.33 ± 6.03 %) and rituximab (64.56 ± 4.35 %) (Supplementary Table 7, Supplementary Fig. 44). These values are comparable to those of trastuzumab (69.82 ± 7.18 %) and cetuximab (66.87 ± 8.14 %), confirming that the platform accommodates antibodies with diverse pI values and structural features.

- Serum stability of mAb-adsorbed GA-lipo formulations

As shown in Supplementary Fig. 45, all four antibody-modified liposomes (N@, R@, T@, C@GA-P1-lipo) maintained stable particle size (generally within 150–180 nm), low PDI (<0.25), and moderate zeta potential (-14 to -5 mV) over 48 h in PBS containing 10 % FBS at 37 °C, indicating good colloidal stability under physiological conditions.

- Antibody retention under plasma exposure

The fluorescence signal of FITC-labeled nimotuzumab and rituximab on GA-lipo remained largely intact after 48 h incubation with 50 % mouse plasma, indicating strong binding and minimal displacement by serum proteins (Supplementary Fig. 46b, d).

- Good batch-to-batch reproducibility

As summarized in Supplementary Table 7, three independently prepared batches of each mAb@GA-P1-lipo formulation showed consistent particle size (CV% all <4 %) and adsorption efficiency (CV% ranging from 5.8 % to 12.1 %). These results confirm the reproducibility and robustness of the platform for antibody surface adsorption.

These findings support the broad applicability and scalable reproducibility of GA-lipo for stable and efficient adsorption of a wide range of monoclonal antibodies.

Supplementary Fig. 44 Adsorption efficiency of GA-P1-lipo toward diverse monoclonal antibodies. Bar chart (mean \pm s.d., $n = 9$) comparing adsorption efficiencies for trastuzumab (T), cetuximab (C), nimotuzumab (N) and rituximab (R).

Supplementary Fig.45 Serum stability of antibody-decorated GA-P1-lipo. Time-dependent (0–48 h, 37 °C, 10% FBS) changes in size, PDI and ζ -potential for N@, R@, T@ and C@GA-P1-lipo (mean \pm s.d., $n = 3$).

Supplementary Fig.46 Plasma retention of adsorbed antibodies. (a,c) Calibration curves for FITC-labelled nimotuzumab and rituximab. (b,d) Fluorescence remaining on GA-P1-lipo after 48 h in 50% mouse plasma (mean \pm s.d., n = 3).

Supplementary Table 7 Characterization and batch reproducibility of different antibody-adsorbed GA-P1-lipo formulations.

	Parameter	Batch 1	Batch 2	Batch 3	Mean \pm SD	CV%
TRA@GA-P1-lipo	Adsorption efficiency (%)	78.47 \pm 1.06 (CV=1.35)	66.91 \pm 1.89 (CV=1.89)	64.10 \pm 5.27 (CV=5.62)	69.82 \pm 7.18	10.29
	Size (nm)	103.2 \pm 1.71 (CV=1.66)	104.9 \pm 5.25 (CV=5.01)	104.9 \pm 0.35 (CV=0.33)	104.4 \pm 2.89	2.77
	PDI	0.043 \pm 0.040	0.051 \pm 0.023	0.046 \pm 0.025	0.047 \pm 0.026	
	Zeta potential (mV)	-14.03 \pm 0.94	-13.9 \pm 0.87	-13.96 \pm 0.66	-13.97 \pm 0.73	
Cet@GA-P1-lipo	Adsorption efficiency (%)	66.15 \pm 2.56 (CV=3.87)	58.31 \pm 3.76 (CV=6.48)	76.14 \pm 2.24 (CV=2.95)	66.87 \pm 8.14	12.1
	Size (nm)	106.0 \pm 1.90 (CV=1.79)	101.56 \pm 1.37 (CV=1.37)	104.3 \pm 3.81 (CV=3.66)	103.9 \pm 2.97	2.86

	PDI	0.060±0.039	0.111±0.062	0.033±0.018	0.068±0.050	
	Zeta potential (mV)	-8.23±1.46	-8.97±0.524	-8.4±0.346	-8.55±0.86	
Nimotuzumab @GA-P1-lipo	Adsorption efficiency (%)	68.01±1.28 (CV=1.63%)	57.66±6.37 (CV=9.42)	67.33±1.69 (CV=2.17)	64.33±6.03	8.11
	Size (nm)	104.1±2.06 (CV=1.98)	104.9±1.10 (CV=1.37)	103.3±1.68 (CV=3.66)	104.2±1.61	1.55
	PDI	0.097±0.074	0.044±0.001	0.053±0.064	0.064±0.054	
	Zeta potential (mV)	-4.57±0.183	-5.27±0.576	-5.47±1.19	-5.10±0.78	
	Adsorption efficiency (%)	67.46±1.50 (CV=1.94)	62.18±5.89 (CV=8.16)	64.05±4.16 (CV=5.62)	64.56±4.35	5.84
Rituximab @GA-P1-lipo	Size (nm)	106.4±2.16 (CV=2.04)	104.9±5.63 (CV=5.37)	104.6±5.05 (CV=4.83)	105.3±4.02	3.82
	PDI	0.085±0.066	0.165±0.081	0.075±0.090	0.146±0.65	
	Zeta potential (mV)	-6.07±0.38	-8.50±0.49	-6.96±1.24	-7.18±1.27	

3) While the authors claim GA-lipo outperforms conventional chemical conjugation, no direct side-by-side comparison is included. Covalent conjugation methods (e.g., maleimide-thiol, click chemistry) are widely used in clinically approved nanomedicines. This study would be more compelling if it provided quantitative performance metrics (e.g., targeting efficiency, stability, in vivo tumour accumulation) compared to covalent antibody-liposome conjugates, as it could allow the industry to adopt such unique approaches.

Response: We agree that a head-to-head comparison is essential. We therefore prepared trastuzumab-decorated liposomes by a standard maleimide–thiol method (T@Mal-lipo) and directly compared them with our physical-adsorption system (T@GA-lipo). Key translational attributes were assessed as follows (see

Supplementary Fig. 32a–f):

- Formulation stability (10% FBS, 37 °C).

Both T@GA-lipo and T@Mal-lipo maintained comparable hydrodynamic size (\approx 120–160 nm), low PDI (<0.2) and slightly negative ζ -potential (-14 to -5 mV) over 48 h, indicating similar colloidal stability (Fig. 32a).

- Cellular uptake under serum challenge.

In HER2-overexpressing SKOV3 cells, T@GA-lipo achieved higher DiO-positive cell percentages than T@Mal-lipo at all serum levels tested. The difference became most pronounced in 50% FBS, where uptake of T@GA-lipo remained $>70\%$, whereas T@Mal-lipo dropped to $<20\%$ (Fig. 32b–c). These data highlight the superior serum-tolerance of the GA-lipo interface.

- In-vivo tumor retention and off-target distribution.

Live imaging of DiR-labelled formulations in SKOV3 xenograft mice showed that both groups accumulated rapidly in tumors, reaching maximum radiance between 4–8 h. Importantly, the signal of T@GA-lipo persisted through 36 h and 48 h, whereas that of T@Mal-lipo declined significantly ($p = 0.0414$ and 0.0006 at 36 h and 48 h, respectively; Fig. 32d–e).

Ex-vivo quantification at 48 h revealed notably lower liver and spleen fluorescence for T@GA-lipo ($p < 0.0001$), with no statistical difference in heart, lung or kidney, confirming reduced RES uptake and enhanced tumor selectivity (Fig. 32f).

Taken together, these results demonstrate that GA-lipo matches the physical stability of covalent conjugates while offering superior serum-resistant cellular uptake, prolonged tumor residence and diminished off-target accumulation. We have incorporated these findings on page. 13–14 of the revised manuscript and provided full datasets in Supplementary Fig. 32.

Supplementary Fig.32 Side-by-side comparison of GA-mediated adsorption and maleimide-thiol covalent conjugation. a, Time-dependent size, PDI and ζ -potential of T@GA-lipo and T@Mal-lipo in PBS + 10% FBS at 37 °C (mean \pm s.d., n = 3). b, SKOV3 uptake of DiO-labelled liposomes after 12 h (0% FBS). c, Uptake at 10% and 50% FBS (12 h) determined by flow cytometry (mean \pm s.d., n = 3). d, In vivo fluorescence imaging of DiR labeled liposomes in tumor at different determined times after injection administration and Ex vivo fluorescence imaging of major organs obtained from the mice at 48 h post-injection. Three biological replicates were measured (n = 3). e, Semi-quantitative tumour radiance over time (mean \pm s.d., n = 3). f, Fluorescence intensities of excised organs at 48 h (mean \pm s.d., n = 3; one-way ANOVA with multiple-comparison post-test).

4) The manuscript states that GA-lipo is stable at 4°C for six weeks, but this does not address long-term storage stability, which is crucial for future clinical translation. Stability in biological fluids (e.g., plasma, whole blood) beyond 24–48

hours has not been tested. An evaluation of the liposome integrity in human plasma over 7+ days to simulate clinical pharmacokinetics. We would also like to see long-term storage stability data at different temperatures (e.g., -20°C, 25°C) over at least three months. Additional DLS and Zeta potential measurements over time would provide insight into nanoparticle aggregation and degradation.

Response: We thank the reviewer for this valuable suggestion regarding the translational stability of the GA-lipo system. In response, we conducted additional studies to evaluate both long-term storage stability and biological fluid stability under clinically relevant conditions.

- Long-term storage stability at 4 °C and 25 °C

The physicochemical stability of GA-DX-lipo and DX-lipo was monitored over a period of 90 days. Dynamic light scattering (DLS) measurements showed no significant changes in particle size, polydispersity index (PDI), or zeta potential at either 4 °C or 25 °C (Supplementary Fig. 35b–c), indicating excellent colloidal stability during storage. Furthermore, Cryo-TEM analysis revealed that GA-DX-lipo retained intact vesicular morphology after 3 months at 25 °C, with no detectable difference from freshly prepared liposomes (Supplementary Fig. 35d). These results suggest that the formulation maintains good physical integrity over an extended period and supports its potential for further shelf-life development.

- Stability in biological fluids over 10 days

To simulate systemic exposure, we evaluated the stability of GA-DX-lipo and T@GA-DX-lipo in 50 % rat plasma at 37 °C for 10 days. As shown in Supplementary Fig. 41, both formulations exhibited minimal variation in size (<12 nm), PDI, and surface charge over the full duration. Notably, the antibody-modified formulation (T@GA-DX-lipo) showed even greater size stability, likely due to enhanced interfacial stabilization conferred by the galloyl-antibody interactions.

Together, these results demonstrate that GA-lipo possesses strong colloidal stability both during extended storage and under prolonged biological exposure, providing encouraging support for future translational development.

Supplementary Fig.35 Release profile and long-term storage stability of DXdd-loaded liposomes. a, Cumulative DXdd release from DXdd solution, DX-lipo, and GA-DX-lipo in PBS at 37 °C. (b, c) Size, PDI, and zeta potential of DX-lipo and GA-DX-lipo stored at (b) 4 °C and (c) 25 °C for 90 days (mean ± s.d., n = 3). d, Cryo-TEM images of GA-DX-lipo: left, freshly prepared; right, after 3 months at 25 °C. Scale bars: 100 nm.

Supplementary Fig.41 Plasma stability of DXdd-loaded formulations. Size, PDI and zeta potential of DX-lipo, GA-DX-lipo, T/DX-lipo and T@GA-DX-lipo in mice plasma at 37 °C for 10 days (mean ± s.d., n = 3).

5) In the supplementary section, the pharmacokinetics (Supplementary Table 4, 5) provides valuable insight into liposome circulation time and tumor accumulation, but lacks long-term clearance data. The biodistribution study (Supplementary Fig.

22, 35) is well-executed, but it would be useful to track organ-specific clearance over time.

Response: We appreciate the reviewer's insightful suggestion regarding clearance profiling. To address this, we extended both the pharmacokinetic analysis and organ-level biodistribution studies to longer timepoints.

- Extended pharmacokinetics up to 72 h.

As shown in Supplementary Fig. 23c and Supplementary Table 4, we expanded the blood sampling window to 72 h. Trf@GA-P0-DOX-lipo and Trf@GA-P1-DOX-lipo exhibited significantly higher AUC(0-t) values (1985.25 ± 63.52 and 2659.80 ± 315.77 $\mu\text{g}\cdot\text{h}/\text{L}$, respectively) compared to non-targeted formulations. This suggests prolonged circulation and enhanced systemic exposure due to effective antibody adsorption.

- Organ-specific biodistribution and clearance trends.

We monitored the biodistribution of DOX (active drug) and DXdd (prodrug) in major organs—including liver, spleen, kidney, heart, lung, and tumor—at 2, 6, 12, 24, and 48 h post-injection (Fig. 2g; Supplementary Figs. 24 and 48). Across all formulations, drug accumulation in normal organs generally peaked between 12–24 h and declined by 48 h, indicating progressive systemic clearance. Compared to non-targeted controls, Trf@GA-lipo formulations exhibited reduced hepatic and splenic uptake, sustained tumor retention with consistently higher drug levels over time, and improved tumor-to-organ ratios at later timepoints, collectively supporting enhanced selective delivery and minimized off-target exposure.

These results underscore the favorable pharmacokinetic and biodistribution profiles of the targeted GA-lipo system, including prolonged blood circulation, efficient tumor accumulation, and steady clearance from off-target organs. Relevant descriptions have been integrated into the revised manuscript on page. 10 and 19.

Supplementary Fig.23 c, Plasma concentration–time profiles of Trf@GA-lipo formulations after intravenous administration (n = 3, mean ± s.d.).

Supplementary Table 4 Pharmacokinetic parameters of DOX solution and various liposomal formulations after intravenous administration (n = 3)

	Cmax (ug/L)	AUC(0-t) (ug/L*h)	t1/2z (h)	CLz (L/h/kg)	Vz (L/kg)
DOX	2.09±0.69	0.69±0.13	0.53±0.25	3.552±0.556	2.701±1.348
DOX-lipo	105.50±4.67	1643.94±160.14	15.51±1.72	0.001±0.001	0.033±0.006
Trf/DOX-lipo	98.09±1.90	1573.78±36.03	12.19±1.49	0.002±0.001	0.028±0.004
GA-P0-DOX-lipo	186.99±7.15	1081.09±138.31	4.63±0.093	0.002±0.001	0.016±0.005
Trf@GA-P0-DOX-lipo	230.18±15.18	1985.25±63.52	7.64±1.37	0.001±0.001	0.014±0.003
GA-P1-DOX-lipo	240.53±22.46	2153.23±207.12	7.69±2.48	0.001±0.001	0.013±0.004
Trf@GA-P1-DOX-lipo	271.40±39.28	2659.80±315.77	11.48±1.11	0.001±0.001	0.015±0.002
GA-P3-DOX-lipo	20.99±3.20	22.41±4.86	1.03±0.62	0.106±0.033	0.141±0.066
Trf@GA-P3-DOX-lipo	25.71±2.10	54.34±13.30	1.77±1.20	0.048±0.012	0.109±0.048

Figure 2g, DOX concentrations in tumors at 2, 6, 12, 24, and 48 h following intravenous injection of Trf@GA-lipo formulations (n = 3, mean ± s.d.).

Supplementary Fig.24 In vivo biodistribution of the 4T1 tumor-bearing mice at 2, 6, 12, 24 and 48 h after i.v. administration to mice with DOX-loaded liposomes at a DOX equivalent dose of 10 mg/kg (n = 3, mean ± s.d.).

Supplementary Fig.48 In vivo biodistribution of the 4T1 tumor-bearing mice at 2, 6, 12, 24 and 48 h after i.v. administration to mice with DXdd-loaded liposomes at a DXdd equivalent dose of 5 mg/kg (n = 3, mean ± s.d.).

6) The cell viability assays (Supplementary Fig. 25a, 25b) demonstrate GA-lipo's targeted delivery, but non-specific uptake by Fc receptor-expressing immune cells is not assessed. (e.g., cytokine response) is not explored. We would like to see control experiments that validate that GA-lipo is receptor-mediated endocytosed

instead of passive endocytosis.

Response: We appreciate the reviewer's insightful suggestion. To determine whether T@GA-lipo is internalized via receptor-mediated endocytosis rather than non-specific Fc receptor (FcR)-mediated uptake, we conducted two complementary studies:

- Non-specific uptake in Fc γ receptor-expressing macrophages

We evaluated the cellular uptake and cytokine response of T@GA-lipo in THP-1-derived tumor-associated macrophages (TAMs), which are known to express high levels of Fc γ receptors (Fc γ R; *Angew. Chem. Int. Ed. Engl.* 2024, 63, e202400538). Flow cytometry analysis showed no increase in T@GA-lipo uptake compared to non-targeted liposomes, indicating minimal Fc γ R interaction (Supplementary Fig. 31a). In parallel, cytokine profiling by ELISA revealed no significant elevation of TNF- α or IL-6, in contrast to the trastuzumab and T/lipo groups containing unadsorbed antibody (Supplementary Fig. 31b–c). IL-10 levels also remained unchanged, suggesting no detectable immunostimulatory effects.

- Receptor-blocking assay in HER2+ cells

To confirm specific targeting, we performed competitive inhibition experiments in HER2-positive SKOV3 cells. Cells were pre-treated with excess anti-HER2 antibody to block receptor binding prior to incubation with T@GA-lipo. Flow cytometry analysis showed significantly reduced cellular uptake under HER2 blockade (Supplementary Fig. 30), confirming that internalization is driven by antibody–antigen interactions rather than passive diffusion.

These results have been included in the revised manuscript (page. 14), supporting that T@GA-lipo uptake is mediated by specific receptor–antibody interactions and does not induce non-specific Fc receptor–related immune responses.

Supplementary Fig.30 a, The influence of blocked T@GA-lipo with BSA on the the uptake of SKOV3 cells. T@GA-lipo was prepared by incubating T@GA-lipo with BSA at 25 °C for 1 h. The incubation DiO concentration was 5 µg/ml. **b**, SKOV3 cells were incubated with DiO-labeled GA-lipo or T@GA-lipo (5 µg/mL, 8 h). To assess receptor specificity, SKOV3 cells were pre-treated with excess anti-HER2 antibody (30 min) to block HER2 receptors prior to T@GA-lipo incubation. Cellular uptake was quantified by flow cytometry and expressed as the percentage of DiO-positive cells. Data are presented as mean ± s.d. from three independent experiments (n = 3). Statistical significance was determined by one-way ANOVA.

Supplementary Fig.31 Evaluation of non-specific uptake and cytokine secretion in Fc receptor-expressing macrophages. THP-1-derived macrophages were used to assess the non-specific uptake and immunostimulatory effects of different liposome formulations. **a**, Flow cytometry analysis of DiO-labeled liposome uptake by THP-1 macrophages. Cells were incubated

with various liposomes (DiO-labeled lipo, GA-lipo, T/lipo, or T@GA-lipo; 5 $\mu\text{g}/\text{mL}$) for 12 h in complete medium. No significant increase in uptake was observed with T@GA-lipo compared to control groups. b-d, Cytokine secretion profiles of THP-1 macrophages after 24 h incubation with trastuzumab (TRA), non-targeted lipo, GA-lipo, T/lipo, or T@GA-lipo. The concentrations of b, TNF- α , c, IL-6, and d, IL-10 were measured by ELISA. Data are shown as mean \pm s.d. from three independent experiments ($n = 3$). Statistical significance was determined by one-way ANOVA; ns, not significant.

7) The in vivo tumor model (Supplementary Fig. 36, 37) confirms therapeutic efficacy, yet systemic immune activation (e.g., cytokine response) is not explored. Perform cytokine analysis (IL-6, TNF- α , IFN- γ) in serum samples from treated mice to check for systemic immune activation or suppression.

Response: We thank the reviewer for this important suggestion. To evaluate potential systemic immune activation following treatment, we performed cytokine analysis using serum samples collected from tumor-bearing mice five days post-administration. Levels of IL-6, TNF- α , and IFN- γ were quantified using high-sensitivity ELISA kits, with three biological replicates per group. All samples were processed under the same conditions to ensure consistency and comparability.

These cytokine profiles have been incorporated into the revised manuscript (page 21), and are presented in Supplementary Fig. 56. The results provide immunological safety data that complement the in vivo therapeutic efficacy findings.

Supplementary Fig.56 Cytokine analysis in serum samples of mice after treatments, including (IL-6, TNF- α , IFN- γ). Three biological replicates were measured ($n = 3$).

8) In Figure 5f, the dosage of GA-lipo administered in the mouse treatment experiment was 5 mg/kg. Please explain the rationale behind selecting this dosage and whether it reflects preclinical optimization or other considerations. Additionally, there is an inconsistency regarding the tumor inoculation method. The figure caption states that SKOV3 cells were inoculated orthotopically, but Figure 5f and the methods section indicate subcutaneous inoculation into the underarm of female BALB/c-nu mice. Please clarify the correct method of tumor inoculation.

Response: The 5 mg/kg dose was selected on the basis of a maximum tolerated dose (MTD) study performed in female BALB/c-nu mice. Free DXd at 10 mg/kg caused acute toxicity and mortality, whereas DXdd-loaded liposomes were non-lethal up to the same dose but produced significant weight loss. Doses of 3 mg/kg and 5 mg/kg were well tolerated, showing no behavioral changes or body-weight reduction (Supplementary Fig. 49). We therefore chose 5 mg/kg as an efficacy dose that balances potency and safety; the corresponding description has been added to the manuscript (page 20-21).

Regarding the tumor model, all SKOV3 cells were implanted subcutaneously into the right axillary (underarm) region of BALB/c-nu mice. The caption of Fig. 5f has been corrected accordingly to remove the inadvertent “orthotopic” reference.

Supplementary Fig.49 Body-weight monitoring after administration of different formulations.

BALB/c-nu mice were intravenously injected with free DXd or DXdd-loaded liposomes at doses of 3, 5, or 10 mg/kg, and body weight was monitored over time (n = 3, mean ± s.d.).

9) It is known that galloyl acid has tumor inhibitory properties. In Figure 5A, the cytotoxicity of GA-DXdd-lipo appears to be higher than that of DXdd-lipo, despite similar release behaviors between the two formulations. The authors should discuss whether the galloyl acid modification contributes directly to the enhanced therapeutic effect observed.

Response: We appreciate the reviewer's suggestion. We agree that the enhanced cytotoxicity of GA-DX-lipo cannot be attributed to differential drug release kinetics, as both formulations exhibit similar release profiles. Based on our mechanistic investigations, we propose that the galloyl acid modification directly contributes to the improved therapeutic efficacy primarily by enhancing cellular uptake, rather than through intrinsic cytotoxic effects of gallic moiety itself. This conclusion is supported by the following experimental evidence.

To evaluate potential intrinsic cytotoxicity of the gallic moiety, we performed additional experiments with galloylated blank liposome GA-lipo and non-galloylated blank liposome (lipo). Results confirmed that neither blank formulation exhibited significant cytotoxicity across the tested concentration range (0.004-10 $\mu\text{g/mL}$ GA-P1-Chol) (Supplementary Fig.47). This definitively rules out direct tumor-inhibitory effects of the GA-modified carrier itself under our experimental conditions. Although there have been many studies that reported the tumor cells inhibitory properties of galloyl acid, this inhibitory effect is concentration-dependent and requires high concentration of galloyl acid. However, the equivalent concentration of galloyl acid we chose was very low for *in vitro* cytotoxicity. Thus, galloyl acid did not directly contribute to the tumor inhibitory effect.

Cellular uptake studies (Fig. 4a-c) revealed that GA-DXd-lipo showed higher intracellular accumulation compared to DXd-lipo in different FBS concentrations. This result was aligned with prior literature that polyphenol surface modifications (e.g., tannic acid) could promote nanoparticle internalization (*J. Mater. Chem. B*, 2022, 10, 1561–1570). The mechanism for enhanced uptake may be attributed to the multivalent

interactions between gallic moiety's phenolic hydroxyl groups and cellular components. Abundant phenolic groups facilitate strong, multivalent H-bonding and Van der Waals Forces (vdWF) interactions with the tumor cell surface (interaction with extracellular matrix and affinity for overexpressed receptors, like EGFR), enhancing adhesion and endocytosis (*Nat Biomed Eng*, 2018, 2, 304–317).

These results and discussions have been included in the revised manuscript (pages 18-19) and Supplementary Fig.47.

Supplementary Fig.47 Cell viability treated with various concentrations of galloylated and non-galloylated blank liposomes in SKOV3 cells after 48 h. (n = 5, mean value ± s.d.).

10) Inspired by Enhertu, a clinically approved ADC, the authors designed T@GA-DX-lipo. However, the advantages of T@GA-DX-lipo over Enhertu are not fully elaborated in the manuscript. It is recommended to include a more detailed comparison between the two platforms, highlighting the unique benefits of T@GA-DX-lipo. Additionally, the discussion section would benefit from an objective analysis of the study's limitations and the potential challenges associated with translating these findings into clinical applications.

Response: We sincerely appreciate the reviewer's insightful suggestions. In response, we have revised the manuscript to (1) provide a detailed comparison between our system (T@GA-DX-lipo) and the clinically approved ADC Enhertu, and (2) include an objective discussion of the limitations and translational challenges of our platform (pages 21–22).

To contextualize our strategy with existing HER2-targeted therapies, we compared

T@GA-DX-lipo with Enhertu (trastuzumab deruxtecan). Both platforms utilize trastuzumab for targeting and a deruxtecan-derived payload, but differ significantly in design and mechanism. Enhertu relies on covalent conjugation, requiring precise drug-to-antibody ratios, complex manufacturing, and potential structural perturbation of the antibody. In contrast, T@GA-DX-lipo enables modular, non-covalent antibody assembly via galloyl–Fc interactions, which preserves antibody conformation and simplifies preparation.

Moreover, ADCs are typically limited to 2–8 payload molecules per antibody, while liposomes can encapsulate thousands of drug molecules. In our system, each trastuzumab molecule delivers ~580 DXdd molecules, greatly enhancing the drug-to-antibody molar ratio and therapeutic potency. Mechanistically, galloyl groups mediate directional antibody adsorption, preserving antigen-binding site orientation even in the presence of protein corona, thereby enhancing tumor-specific delivery and synergy.

Despite these promising findings, several limitations should be considered. First, although the galloyl-mediated adsorption strategy offers a convenient route for antibody functionalization, its non-covalent nature may lead to antibody desorption under dynamic *in vivo* conditions. While our *in vivo* data support effective targeting within the experimental window, long-term stability of antibody binding remains to be fully validated. Second, compared to small-molecule antibody–drug conjugates (ADCs) such as Enhertu, the nanoparticle-based system may exhibit distinct intertumoral distribution patterns due to differences in size, diffusivity, and tumor penetration behavior. These differences may influence drug release kinetics and therapeutic outcomes and merit further comparative evaluation. Finally, the translational potential of this platform requires additional validation in large-animal models, including assessments of long-term safety, immunogenicity, and pharmacokinetics under clinically relevant conditions.

Together, this work introduces a generalizable strategy for constructing targeted nanocarriers via non-covalent protein assembly, bridging the precision of ADCs and the flexibility of nanomedicine to support translational advancement in targeted chemotherapy.

Reviewer #2 (Remarks to the Author):

Li et al. presents an interesting surface modification of liposomes via incorporation of gallic acid-modified lipids, allowing stable noncovalent binding of antibodies to the surfaces of the nanoparticles. The manuscript is overall clearly written and the therapeutic results are convincing in a cancer model of Dox delivery. Publication is recommended after some modifications/additions. Mainly, additional details are needed for methods and figures, as well as additional methods of characterizing protein adsorption to the liposome surfaces. Suggested changes are outlined below:

Major issues:

1) Overall, experimental details need to be described in more detail in the results and in the figure captions. In many cases, the route of administration nor concentrations of administered agents is described. These should be clearly stated in both locations so that the reader can better interpret the data.

Response: Thanks for the reviewer's valuable advice. We have added more detailed description of the experiments and figure captions in the revised manuscript. The revised portions have been marked in red.

2) The experiments characterizing the orientation of antibodies are not convincing and thus do not support their conclusion that the Fc region of IgG1 primarily complexes with galloyl moieties on GA-P1-lipo. Their experiments assume only two possible orientations of the antibody, while numerous are possible that could account for difficulty in the anti-Fc antibodies binding. There could be diverse denatured structures in between. Alternative methods of characterization/validation are necessary or their conclusions should be updated in the text to be less confident.

Response: We appreciate the reviewer's insightful comment and fully agree that antibody orientation is a complex phenomenon with multiple potential configurations beyond the two idealized states initially considered. To address this concern, we conducted molecular dynamics (MD) simulations to gain deeper insight into the

binding behavior between galloyl moieties and different regions of trastuzumab. The revised text appears in the manuscript on pages 12–13 and Figure 3g–j and Supplementary Figures 27–28.

As shown in Figure 3g–j and Supplementary Figures 27–28, the simulations revealed that galloyl groups (e.g., GA-cholesterol) tend to associate preferentially with the Fc region of trastuzumab. This was supported by detailed analysis of intermolecular forces, which included hydrophobic interactions, hydrogen bonding, and electrostatic attractions. Notably, the total average binding energy for the Fc region was consistently higher than that for the Fab region, suggesting stronger affinity and more stable complexation at the Fc interface.

While these results support our initial hypothesis, we acknowledge that the *in vivo* or *in situ* orientation of antibodies may still be influenced by protein conformation, lipid microenvironments, or serum protein interactions. Therefore, we have revised the manuscript to present our conclusions with greater caution and to emphasize that the MD simulations suggest a preferential, rather than exclusive, Fc-region binding.

Supplementary Fig.27 Conformational stability of trastuzumab during MD simulation. a, Time-evolution of backbone RMSD (root-mean-square deviation) for the full antibody (black) and its sub-domains (Fab1, Fab2, hinge, Fc). b, Solvent-accessible surface area (SASA) of the full antibody as a function of time.

Supplementary Fig.28 Molecular interaction analysis between trastuzumab and GA-P1-Chol based on molecular dynamics simulations. a, Overall docking model of trastuzumab and GA-P1-Chol, showing the spatial location of eight identified binding domains (domains 1-8) across Fab1, Fab2, and Fc regions. GA-P1-Chol is shown in magenta; individual antibody chains are colored distinctly. b, Representative close-up views of GA-P1-Chol interactions within each domain. Key residues involved in hydrogen bonding (yellow dashed lines) or hydrophobic interactions are labeled. Domain1 (D658, S660): 5 hydrogen bonds; Domain2: hydrophobic interactions; Domain3 (H692, L413): 2 hydrogen bonds; Domain4 (D633, S632, A501): 4 hydrogen bonds; Domain5 (E434): 2 hydrogen bonds; Domain6: hydrophobic interactions; Domain7 (R20): 1 hydrogen bond; Domain8 (T301, D244): 2 hydrogen bonds. c, Binding free energy decomposition of GA-P1-Chol to each domain based on per-residue energy contribution analysis. Bar graphs show the energy contribution (kcal/mol) of key residues; negative values indicate favorable interactions.

Figure 3 g–j. Molecular-dynamics analysis of GA-P1-Chol binding to trastuzumab. g, Simulation snapshot of an equilibrated TRA protein bound to twenty GA-P1-Chol molecules (in purple). The ligand–protein interfaces are stabilized by hydrophobic pockets and hydrogen-bond networks, as indicated from domain 1 to domain 8, maintaining the structural integrity of TRA. h, Time evolution of hydrogen bonds formed between GA-P1-Chol and the full-length trastuzumab (black) or its Fc region (red) during a 100 ns MD simulation. The increasing trend indicates progressive stabilization of the complex over time, with a higher number of hydrogen bonds observed in the full antibody structure. i, Decomposition of binding energy contributions between GA-P1-Chol and each domain (1–8), showing Coulombic (electrostatic) and Lennard-Jones (van der Waals) interactions. j, Time-resolved total interaction energy (kcal/mol) between GA-P1-Chol and individual antibody domains during the last 20 ns of MD simulation.

3) Overall, there is insufficient characterization of what is binding to the nanoparticle surfaces, their structure, and their stability. It was odd that a 70% binding efficiency was achieved regardless of the protein concentration in the solution during loading. Are all accessible spaces bound? Many noncovalent

techniques for protein binding use a “filler/blocking” protein to cover locations that are not bound by the desired antibodies. Is this necessary here? Do proteins from serum subsequently bind to the nanoparticle surfaces if these additional locations are not blocked? Does the amount of incorporated gallic acid modified lipids correlate with the amount of bound protein?

Response: We thank the reviewer for the insightful comment. We agree that the initial observation of ~70% protein binding efficiency regardless of the protein concentration raised concerns about the validity of the quantification. Upon further investigation, we found that this anomaly occurred only in our early-stage experiments (Supplementary Fig.15 and Figure 2b) where the FITC-labeled protein was purified by dialysis. At that time, incomplete removal of unbound FITC might have led to imprecise of protein binding efficiency due to interference in the fluorescence-based quantification. To address this, we subsequently implemented a more rigorous purification protocol using Sephadex-based size exclusion chromatography (SEC) to completely remove free FITC before the labeling protein was used in any binding experiments. All subsequent experiments and data shown in the revised manuscript were obtained using purified FITC-protein. To ensure data integrity, we have re-measured the key representative data points from the early experiments using the purified protein (Figure 2b, Supplementary Fig.11).

Additionally, we evaluated GA-P0-lipid incorporation (1%, 5%, 10%) and found transferrin adsorption increased with GA content, with 10% yielding the highest binding (Supplementary Fig.11d). Formulations above 10% GA-P0-lipid were unstable. These results indicate that protein adsorption is GA-content-dependent. Relevant revisions have been added to the manuscript pages 7.

We investigated whether blocking unbound surface sites with proteins like BSA is necessary for T@GA-lipo. Testing this approach revealed that BSA blocking had no significant impact on cellular uptake, indicating that it is not required in our system (Supplementary Fig.30a). Although serum proteins may adsorb to unblocked surfaces, their presence does not appear to interfere with the targeting function. Relevant

revisions have been added to the revised manuscript pages 13.

Figure 2b Super centrifugation was used to detect the adsorbed antibodies on the surface of GA-lipo. pH value influenced the adsorption ability of polyphenol on the surface of GA-lipo and 7.4 is the preferable pH for various proteins. (pI: Bovine Serum Albumin, BSA 4.7, transferrin, TRF 5.4, Human hemoglobin, HBB, 6.8, Trastuzumab, TRA 8.7, Trypsin 10.1) Data are represented as the MFI of FITC-positive nanoparticles. (n = 3, mean value \pm s.d.).

Supplementary Fig.11 The influence of incubating conditions on Trf adsorption and uptake *in vitro*. **a**, Incubating time and **b**, Temperature of Trf and GA-lipo influence the adsorption efficiency of Trf. The mean \pm s.d. is displayed from three parallel experiments (n = 3). **c**, The influence of Trf concentration and **d**, modified GA-Chol on the absorption efficiency of protein. The mean \pm s.d. is displayed from three parallel experiments (n = 3).

Supplementary Fig.30a a, The influence of blocked T@GA-lipo with BSA on the the uptake of SKOV3 cells. T@GA-lipo was prepared by incubating T@GA-lipo with BSA at 25 °C for 1 h. The incubation DiO concentration was 5 µg/ml. The mean ± s.d. is displayed from three parallel experiments (n = 3).

4) Circular dichroism analysis could help assess changes in secondary structure upon binding of proteins and antibodies. It is not clear if denaturation occurs at the nanoparticle surface.

Response: We thank the reviewer for the insightful suggestion to evaluate potential structural changes of proteins upon surface adsorption using circular dichroism (CD). In response, we performed both CD spectroscopy and Fourier-transform infrared (FTIR) analysis on transferrin (Trf) and trastuzumab (TRA) adsorbed onto GA-lipo to assess whether protein denaturation or significant secondary structural changes occurred. Relevant revisions have been added to the manuscript pages 8 and 11.

For transferrin, FTIR spectra of Trf@GA-lipo revealed shifts in the characteristic amide I (~1640 cm⁻¹) and amide II (~1543 cm⁻¹) bands, suggesting specific interactions between the protein backbone (-NH₂ groups) and the galloyl moieties on the GA-lipo surface (Supplementary Fig. 19c). However, the CD spectra of Trf@GA-lipo remained nearly unchanged compared to free Trf (Supplementary Fig. 19d), indicating preservation of its secondary structure upon adsorption.

Similarly, for trastuzumab, FTIR analysis showed comparable band shifts, while

CD spectra of T@GA-lipo were nearly identical to those of free TRA (Supplementary Fig. 26c–d), suggesting that the adsorption process did not disrupt the antibody’s secondary structure.

Taken together, these results suggest that while non-covalent interactions may induce minor conformational adjustments, the GA-lipo platform preserves the structural integrity of adsorbed therapeutic proteins and antibodies, supporting its suitability for functional delivery applications.

Supplementary Fig.19b-c Secondary structure of transferrin adsorbed on GA-lipo. b, FT-IR spectra and c, CD spectra of Trf and Trf@GA-lipo.

Supplementary Fig.26c-d Secondary structure of TRA adsorbed on GA-lipo. c, FT-IR spectra and d, CD spectra of TRA and T@GA-lipo.

5) After antibody binding, what proteins can subsequently bind to the nanoparticle surfaces upon exposure to serum? Do proteins exchange over time (i.e. Vroman effect)?

Response: We appreciate the reviewer’s thoughtful question regarding protein exchange dynamics on the nanoparticle surface following antibody adsorption,

particularly in the context of the Vroman effect. These results and discussions have been included in the revised manuscript (pages 17):

To investigate the stability of antibody binding, we conducted isothermal titration calorimetry (ITC) to quantify the interaction between GA-lipo and trastuzumab. As shown in Supplementary Fig. 39a–b and Table 6, the dissociation constant (KD) was determined to be 0.329 μ M, indicating a strong binding affinity and supporting the formation of a stable antibody corona on the nanoparticle surface.

To further assess potential protein exchange upon exposure to serum, we performed SDS-PAGE analysis of TRA@GA-lipo after incubation with plasma at 37 °C for varying durations (5 min to 24 h). As shown in Supplementary Fig. 39c, while the total protein composition on the liposome surface evolved over time—suggesting dynamic adsorption of serum proteins—the intensity of the trastuzumab band remained largely unchanged. This result indicates that trastuzumab remains stably adsorbed on GA-lipo despite the presence of competing serum proteins.

Supplementary Table 6 ITC-derived thermodynamic parameters for the binding of trastuzumab to GA-lipo.

Titrant	Titrand	KD(M)	N	Δ H(KJ/mol)	Δ S(J/mol·K)
GA-lipo	trastuzumab	3.29e-07	1.83	-38.55	-5.25

Supplementary Fig.39 Characterization of protein binding and corona formation on T@GA-lipo.

a, Isothermal titration calorimetry (ITC) thermogram and b, scatter plot showing the interaction between trastuzumab and GA-lipo. c, Time-dependent formation of the protein corona on T@GA-lipo nanoparticles in mouse plasma. After incubation with plasma for various time points (5 min to 24 h), protein-bound nanoparticles were isolated and analyzed by SDS-PAGE. TRA: free trastuzumab as control.

Minor issues:

6) Several grammatical errors were found throughout the text. Thorough reread and editing is recommended

Response: We are sorry for our carelessness. We have checked the grammatical errors throughout the manuscript. The revised portions have been highlighted in red.

7) Many of the figures, especially in the supplement are a) not referenced in the text or b) referenced out of order. For example, Supplementary figs 9-12 are not discussed and 13a and 13b are out of order.

Response: We agree with the reviewer's comment. We have added relevant description in the revised manuscript for the figures not referenced in the text. Moreover, the order of figures has been adjusted to ensure appearing as sequence in the text.

Manuscript ID: NCOMMS-25-02797-T

Title: Galloylated Liposomes Overcoming Protein Corona Shielding for Enhanced Targeted Drug Delivery

Dear reviewers,

We would like to express our sincere gratitude for your kind comments and the time you dedicated to reviewing our manuscript. Your constructive feedback has been invaluable in improving the quality of our work.

We appreciate your positive evaluation and have revised the manuscript carefully in accordance with the editorial policies and formatting requirements. We hope that the updated version now meets the standards for publication.

Your sincerely

Yongjun Wang, Ph.D.

Response to Reviewers' comments:

Reviewer #1

Comments: the authors have addressed my concerns and the manuscript is now acceptable for publication.

Response: We feel great thanks for your positive comments and valuable suggestions to improve the quality of our manuscript.

Reviewer #2

Comments: Authors have thoroughly addressed all issues that were raised and provided interesting and supportive new data that improves the manuscript. Publication of the manuscript in its current form is recommended.

Response: We sincerely appreciate your positive comments and professional review work on our article.